# Non-asymptotic Error Bounds in $\mathcal{W}_2$-Distance with Sqrt(d) Dimension Dependence and First Order Convergence for Langevin Monte Carlo beyond Log-Concavity

**Bin Yang** [*1]  **Xiaojie Wang** [*1]

## Abstract

Generating samples from a high dimensional probability distribution is a fundamental task with wide-ranging applications in the area of scientific computing, statistics and machine learning. This article revisits the popular Langevin Monte Carlo (LMC) sampling algorithms and provides a non-asymptotic error analysis in $\mathcal{W}_2$-distance in a non-convex setting. In particular, we prove an error bound $O(\sqrt{d}h)$, which guarantees a mixing time $\tilde{O}(\sqrt{d}\epsilon^{-1})$ to achieve the accuracy tolerance $\epsilon$, under certain log-smooth conditions and the assumption that the target distribution satisfies a log-Sobolev inequality, as opposed to the strongly log-concave condition used in (Li et al., 2019; 2022). This bound matches the best one in the strongly log-concave case and improves upon the best-known convergence rates in non-convex settings. To prove it, we establish a new framework of uniform-in-time convergence for discretizations of SDEs. Distinct from (Li et al., 2019; 2022), we start from the finite-time mean-square fundamental convergence theorem, which combined with uniform-in-time moment bounds of LMC and the exponential ergodicity of SDEs in the non-convex setting gives the desired uniform-in-time convergence. Our framework also applies to the case when the gradient of the potential $U$ is non-globally Lipschitz with superlinear growth, for which modified LMC samplers are proposed and analyzed, with a non-asymptotic error bound in $\mathcal{W}_2$-distance obtained. Numerical experiments corroborate the theoretical analysis.

---

[*]Equal contribution [1]School of Mathematics and Statistics, Hunan Research Center of the Basic Discipline for Analytical Mathematics, HNP-LAMA, Central South University, Changsha 410083, China. Correspondence to: Xiaojie Wang <x.j.wang7@csu.edu.cn, x.j.wang7@gmail.com>.

*Proceedings of the 42$^{nd}$ International Conference on Machine Learning*, Vancouver, Canada. PMLR 267, 2025. Copyright 2025 by the author(s).

## 1. Introduction

Sampling from a complex, high dimensional target probability measure $\pi(\mathrm{d}x) \propto e^{-U(x)}\mathrm{d}x$, where $U(\cdot): \mathbb{R}^d \to \mathbb{R}$ is a potential function, finds wide-ranging applications in the area of scientific computing, statistics and machine learning (Durmus & Moulines, 2019; Song & Ermon, 2019; Wibisono, 2018; Liang & Su, 2019; Kakade, 2003; Neal, 1992; Welling & Teh, 2011). The Langevin stochastic differential equation (SDE) associated with the target probability measure $\pi$ is defined by:

$$\mathrm{d}X_t = -\nabla U(X_t)\,\mathrm{d}t + \sqrt{2}\,\mathrm{d}W_t, \quad X_0 = x_0,\ t > 0, \quad (1)$$

where $W_{\cdot} := \left(W_{\cdot}^1, W_{\cdot}^2, \cdots, W_{\cdot}^d\right)^T : [0, \infty) \times \Omega \to \mathbb{R}^d$ is $d$-dimensional Brownian motion defined on the filtered probability space $(\Omega, \mathcal{F}, \{\mathcal{F}_t\}_{t\geq 0}, \mathbb{P})$, satisfying the usual conditions. The initial data $x_0 \colon \Omega \to \mathbb{R}^d$ is assumed to be $\mathcal{F}_0$-measurable. Under mild conditions, the Langevin SDE admits $\pi$ as its unique invariant distribution (see, e.g., (Pavliotis, 2014)). However, sampling from the exact dynamics (1) directly is usually not accessible and the numerical discretization becomes a practical option. The popular Langevin sampler is based on the Euler–Maruyama (EM) discretization of (1), given by

$$\bar{Y}_{n+1} = \bar{Y}_n - \nabla U(\bar{Y}_n)h + \sqrt{2h}\zeta_{n+1}, \quad \bar{Y}_0 = x_0, \quad (2)$$

where $h > 0$ is the uniform timestep and $\zeta_k := (\zeta_k^1, \zeta_k^2, \cdots, \zeta_k^d)^T$, $k \in \mathbb{N}$, are i.i.d standard $d$-dimensional Gaussian vectors. By combining gradient information with noise, this algorithm, termed as Langevin Monte Carlo (LMC), allows faster convergence and more efficient exploration of complex, high-dimensional distributions compared to the naive Markov Chain Monte Carlo (MCMC). The capability to generate high quality samples and improve model training makes it popular in many applications such as generative models.

**Related literature.** The study of LMC sampling algorithms has attracted considerable attention over recent years. A central problem is to determine the number of iteration steps (termed as mixing time) guaranteeing that the distribution of the Markov chain is a good approximation of the target

measure $\pi$. To obtain the mixing time, the non-asymptotic sampling error analysis is critical and has been extensively studied in the literature (Cheng & Bartlett, 2018; Dalalyan, 2017a; Durmus et al., 2019; Dalalyan, 2017b; Dalalyan & Karagulyan, 2019; Durmus & Moulines, 2019; Li et al., 2019; 2022; Sabanis & Zhang, 2019; Xu et al., 2018), particularly under a strongly log-concave condition ($m > 0$):

$$\langle x-y, \nabla U\left(x\right)-\nabla U\left(y\right)\rangle \geq m|x-y|^2, \quad \forall x, y \in \mathbb{R}^d. \quad (3)$$

Under this condition as well as the gradient Lipschitzness, the seminal work of Dalalyan (Dalalyan, 2017b) explicitly showed non-asymptotic convergence of order $O(\sqrt{dh})$ to Langevin diffusion with warm start, implying that the LMC achieves $\epsilon$ error, in total variation distance, in $\tilde{O}(\frac{d}{\epsilon^2})$ steps. The authors of (Durmus & Moulines, 2017) improved and extended the results of (Dalalyan, 2017b). In $\mathcal{W}_2$-distance, Dalalyan (Dalalyan, 2017a), Durmus et al (Durmus et al., 2019), Cheng and Bartlett (Cheng & Bartlett, 2018) obtained similar non-asymptotic error bounds. By additionally imposing the Hessian Lipschitz condition on $U$, Durmus and Moulines (Durmus & Moulines, 2019) provided improved non-asymptotic error bound $O(dh)$ in $\mathcal{W}_2$-distance for LMC, which guarantees a mixing time $\tilde{O}(\frac{d}{\epsilon})$. Motivated by a recent work (Chewi et al., 2021) that shows better dimension dependence for a Metropolis-Adjusted improvement of LMC, Li et al (Li et al., 2022) obtained an error bound $O(\sqrt{dh})$ in $\mathcal{W}_2$-distance, under the strongly log-concave condition (3) and additional smoothness conditions on $U$. This leads to $\tilde{O}(\frac{\sqrt{d}}{\epsilon})$ mixing time bound, which turns out to be optimal (in terms of order) for LMC in both dimension $d$ and accuracy tolerance $\epsilon$.

Nevertheless, the strongly log-concave condition is seldom satisfied in practice. An interesting and natural question thus arises:

**(Q).** *Can the error bound $O(\sqrt{d}h)$ in $\mathcal{W}_2$-distance still hold true for LMC without the strongly log-concave condition?*

Before answering this question, we highlight that the dynamics of the Langevin SDE (1) in non-convex settings is crucial but non-trivial (see, e.g., (Eberle, 2016; Luo & Wang, 2016; Wang, 2020) and references therein). Following the idea of the reflection couplings, Eberle (Eberle, 2016) showed contraction in $\mathcal{W}_1$-distance for Langevin SDE (1) with the potential $U$ being strongly convex in the long distance. When the unique invariant probability measure satisfies the log-Sobolev inequality (LSI), the author of (Wang, 2020) provided $\mathcal{W}_2$-convergence to the equilibrium of the Langevin SDE (1).

Recently, there has been some progress in the non-asymptotic error analysis for (modified) LMC, without the aforementioned strongly log-concave condition, see, e.g., (Cheng et al., 2018; Chewi et al., 2024; Mou et al., 2022; Mousavi-Hosseini et al., 2023; Pang et al., 2025; Li et al.,

2025; Pagès & Panloup, 2023; Majka et al., 2020; Neufeld et al., 2025; Vempala & Wibisono, 2019; Lytras & Sabanis, 2025; Lytras & Mertikopoulos, 2024; Li & Wang, 2025). On the condition that the potential is strongly-convex outside a ball but possibly nonconvex inside this ball, Cheng et al (Cheng et al., 2018) established upper bound $\tilde{O}(d\epsilon^{-2})$ on the number of steps (mixing time) required for LMC (2) to achieve the accuracy tolerance $\epsilon$ in $\mathcal{W}_1$-distance. Under convexity at infinity condition, Majka et al. (Majka et al., 2020) showed error bounds $O(d^{1/4}h^{1/4})$ and $O(d^{1/2}h^{1/2})$ in $\mathcal{W}_2$ and $\mathcal{W}_1$-distance, respectively, which guarantees mixing times $\tilde{O}(d\epsilon^{-4})$ and $\tilde{O}(d\epsilon^{-2})$ to achieve the accuracy tolerance $\epsilon$. Under certain log-smooth conditions and the assumption that the target distribution satisfies a log-Sobolev inequality, Mou et al (Mou et al., 2022) obtained an improved Kullback-Leibler divergence bound, implying mixing times of $\tilde{O}(d\epsilon^{-1})$ in total variation distance and $\mathcal{W}_2$-distance. For some non-convex conditions, Pages and Panloup (Pagès & Panloup, 2023) also proved some error bounds for LMC, but does not explicitly specify the dimension dependence. When the drift coefficient is non-globally Lipschitz continuous, (Li et al., 2025; Pang et al., 2025; Neufeld et al., 2025) examined non-asymptotic error analysis of (modified) LMC sampling algorithms under convexity at infinity condition.

**Our contributions.** In this work, we aim to answer the aforementioned question **(Q)** to the positive and show an error bound $\sqrt{d}h$ in $\mathcal{W}_2$-distance for LMC (2), under certain log-smooth conditions and the assumption that the target distribution satisfies a log-Sobolev inequality (see Theorem 2.10), as opposed to the strongly log-concave condition used in (Li et al., 2019; 2022). This bound guarantees a mixing time $\tilde{O}(\sqrt{d}\epsilon^{-1})$, which matches the best one in the strongly log-concave case and improves upon the best-known convergence rates in non-convex settings. To achieve it, we establish a new, non-convex theoretical framework of uniform-in-time convergence for discretizations of general SDEs (see Theorem 3.4). Distinct from (Li et al., 2019; 2022), we start from the finite-time mean-square fundamental convergence theorem, which combined with uniform-in-time moment bounds of LMC and the exponential ergodicity of SDEs in the non-convex setting gives the desired uniform-in-time convergence. Our framework also applies to the case when the gradient of the potential $U$ is non-globally Lipschitz with superlinear growth, for which modified LMC samplers are proposed and analyzed, with a non-asymptotic error bound in $\mathcal{W}_2$-distance obtained (see subsection 2.3). The main contribution of this work can be summarized as follows:

- Based on the finite-time convergence theorem and the exponential ergodicity of SDEs, we establish a new, non-convex theoretical framework of uniform-in-time convergence (in $\mathcal{W}_2$-distance) for discretizations of general SDEs, without requiring the strongly log-

concave condition. The uniform-in-time convergence rate is explicitly expressed in terms of the contraction rate of SDEs and local convergence rates in finite-time of the numerical scheme.

- As one application of the new framework, we prove an error bound $O(\sqrt{d}h)$ in $\mathcal{W}_2$-distance for LMC, guaranteeing a mixing time bound $\tilde{O}(\sqrt{d}\epsilon^{-1})$ to achieve the accuracy tolerance $\epsilon$, under certain log-smooth conditions and the assumption that the target distribution satisfies a log-Sobolev inequality. This bound matches the best one in the strongly log-concave case and improves upon the best-known convergence rates in non-convex settings.

- As another application of the framework, we also consider the case when the gradient of the potential $U$ is non-globally Lipschitz with superlinear growth. For this case, modified LMC samplers are proposed and analyzed, with a non-asymptotic error bound in $\mathcal{W}_2$-distance explicitly revealed.

The structure of this paper is as follows. Section 2 presents main results for LMC sampling algorithms, with an overview of proof also given. In Section 3, a new non-convex framework of uniform-in-time convergence theorem is established, based on which the main results are derived. Several numerical examples are reported in Section 4 and some concluding remarks are given in the last section.

## 2. Main Results

This section presents notation, assumptions and main results for Langevin Monte Carlo (LMC) associated to gradient Lipschitz potential and modified Langevin Monte-Carlo (mLMC) associated to gradient super-linearly growing potential. Also, an overview of proof is given.

**Notation** Throughout this paper, we use $\mathbb{N}$ to denote the set of all positive integers and let $\mathbb{N}_0 := \mathbb{N} \cup \{0\}$. For all $n \in \mathbb{N}$, let $[n] := \{1, 2, \cdots, n\}$ and $[n]_0 := \{0, 1, \cdots, n\}$. For convention, we set $0^0 = 1$. The symbols $\wedge$ and $\vee$ mean "minimum" and "maximum", respectively. We write $\tilde{O}(\cdot)$ to mean that $O(\cdot) \log^{O(1)}(\cdot)$. We also use the notation $\langle \cdot, \cdot \rangle$ and $|\cdot|$ to denote the inner product and Euclidean norm of vectors in $\mathbb{R}^d$, respectively. Let $\|\cdot\|$ and $\|\cdot\|_F$ denote the operator and trace norm of matrices, respectively. For a function $f : \mathbb{R}^d \to \mathbb{R}$, we write $\partial_i f$ to denote the $i$-th partial derivative of $f$. The gradient $\nabla f$ is the vector of partial derivatives $(\partial_1 f, \cdots, \partial_d f)$ and the Hessian $\nabla^2 f$ is the matrix $(\partial^2_{ij} f)_{i,j \in [d]}$. The Laplacian of $f$ is denoted by $\Delta f := tr \nabla^2 f = \sum_{i=1}^d \partial^2_{ii} f$.

Let $\mathcal{B}(\mathbb{R}^d)$ be the Borel $\sigma$-field of $\mathbb{R}^d$ and $\mathcal{P}(\mathbb{R}^d)$ be the space of all probability distributions on $(\mathbb{R}^d, \mathcal{B}(\mathbb{R}^d))$. For two probability measures $\nu_1, \nu_2 \in \mathcal{P}(\mathbb{R}^d)$ we define a coupling (or transference plan) $\varrho$ between $\nu_1$ and $\nu_2$ as a probability measure on $(\mathbb{R}^d \times \mathbb{R}^d, \mathcal{B}(\mathbb{R}^d \times \mathbb{R}^d))$ such that $\varrho(A \times \mathbb{R}^d) = \nu_1(A)$ and $\varrho(\mathbb{R}^d \times A) = \nu_2(A)$ for all $A \in \mathcal{B}(\mathbb{R}^d)$. We then denote by $\Gamma(\nu_1, \nu_2)$ the set of all such couplings and define the $L^p$-Wasserstein distance ($\mathcal{W}_p$-distance in short) between a pair of probability measures $\nu_1$ and $\nu_2$ as

$$\mathcal{W}_p(\nu_1, \nu_2) := \inf_{\varrho \in \Gamma(\nu_1, \nu_2)} \left( \int_{\mathbb{R}^d \times \mathbb{R}^d} |x - y|^p \, \mathrm{d}\gamma(x, y) \right)^{1/p}$$

for $p \geq 1$. Given a filtered probability space $(\Omega, \mathcal{F}, \{\mathcal{F}_t\}_{t \geq 0}, \mathbb{P})$, we use $\mathbb{E}$ to mean the expectation. In addition, denote by $C_b(\mathbb{R}^d)$ (resp. $B_b(\mathbb{R}^d)$) the Banach space of all uniformly continuous differentiable and bounded mappings (resp. Borel bounded mappings). For $l \in \mathbb{N}$, let $C_b^l(\mathbb{R}^d)$ be the subspace of $C_b(\mathbb{R}^d)$ consisting of all $l$-times continuously differentiable functions with bounded partial derivatives. For any $\phi \in C_b(\mathbb{R}^d)$ and $\nu \in \mathcal{P}(\mathbb{R}^d)$, we denote $\nu(\phi) := \int_{\mathbb{R}^d} \phi(x)\nu(dx)$.

### 2.1. Exponential Ergodicity of the Langevin SDE

In this subsection, we first prove the uniform-in-time moment bounds under the dissipativity condition and then provide the exponential convergence in $\mathcal{W}_2$-distance under one-sided Lipschitz condition and the log-Sobolev inequality.

**Assumption 2.1** (Dissipativity condition)**.** There exist two positive constants $\mu$ and $\mu'$, independent of $d$, such that

$$\langle x, \nabla U(x) \rangle \geq \mu|x|^2 - \mu'd, \quad \forall x \in \mathbb{R}^d. \qquad (4)$$

We highlight that all constants used here (i.e., $\mu, \mu'$) and throughout this section do not depend on the dimensions $d$. Instead of a strong convex condition, we put a one-sided Lipschitz condition on $-\nabla U$.

**Assumption 2.2** (One-sided Lipschitz condition)**.** There exists a dimension-independent constant $L > 0$ such that for all $x, y \in \mathbb{R}^d$,

$$\langle x - y, \nabla U(x) - \nabla U(y) \rangle \geq -L|x - y|^2. \qquad (5)$$

**Assumption 2.3** (Log-Sobolev inequality)**.** Let $\{p_t\}_{t \geq 0}$ be the semigroups associated to Langevin SDE (1) admitting a unique invariant distribution $\pi$. Assume that the invariant distribution satisfies the log-Sobolev inequality, namely, there exists a constant $\rho$, independent of $d$, such that, for any $\phi \in C_b^1(\mathbb{R}^d)$,

$$\pi(\phi^2 \log \phi^2) \leq \rho\pi(|\nabla \phi|^2), \quad \pi(\phi^2) = 1. \qquad (6)$$

We mention that, beyond log-concavity, the log-Sobolev inequality (LSI) is a widely used assumption for the target

distribution of interest in the field of Langevin sampling (Vempala & Wibisono, 2019; Lytras & Sabanis, 2025; Mou et al., 2022; Lytras & Mertikopoulos, 2024; Chewi et al., 2024). As indicated by Appendix A in supplementary material to (Mou et al., 2022), strongly convex outside a ball implies LSI. As a consequence, two typical examples satisfying LSI are the Gaussian mixture and double-well potential (see Section 4). For more details on LSI, one can also consult (Ledoux, 2006).

**Lemma 2.4** (Uniform-in-time moment estimate for Langevin SDE). *Let Assumption 2.1 hold and let $\{X_t\}_{t\geq 0}$ be the solution of the Langevin SDE (1). Then there exists a constant $c \in (0, 2\mu)$, independent of $d, t$, such that it holds for all $p \in [1, +\infty)$,*

$$\mathbb{E}\big[|X_t|^{2p}\big] \leq e^{-cpt}\mathbb{E}\big[|x_0|^{2p}\big] + \frac{2(2p-1+\mu')^p}{cp}\Big(\frac{2p-2}{(2\mu-c)p}\Big)^{p-1}d^p. \tag{7}$$

The proof of this lemma can be found in Appendix A. Let $\{p_t\}_{t\geq 0}$ be the Markov semigroup associated to the solution $\{X_t\}_{t\geq 0}$ of SDE (1). Then for any $\nu \in \mathcal{P}(\mathbb{R}^d)$, $\nu p_t$ denotes the distribution law of $X_t$ starting from $X_0 = x_0 \sim \nu$, i.e., $\nu p_t := \mathcal{L}(X_t)$, $t \geq 0$. Next we present a proposition on exponential ergodicity in $\mathcal{W}_2$-distance of the Langevin SDE (1) under LSI.

**Proposition 2.5** (Exponential ergodicity in $\mathcal{W}_2$-distance). *Let Assumptions 2.2 and 2.3 hold. Then for any $t \geq 0$ and initial distribution $\nu = \mathcal{L}(x_0)$, there exist two constants $\mathcal{K}, \eta > 0$, independent of $d, t$, such that the semigroup $p_t$ associated to SDE (1) and its invariant distribution $\pi$ satisfy*

$$\mathcal{W}_2(\nu p_t, \pi) \leq \mathcal{K}e^{-\eta t}\mathcal{W}_2(\nu, \pi). \tag{8}$$

Such an assertion follows from Theorem 2.1 (2) and Theorem 2.6 (2) in (Wang, 2020), where the parameter dependence was not explicitly provided. Noting that the Langevin SDE (1) is driven by additive noise (non-degenerate) and owing to the one-sided Lipschitz condition (5) and LSI (6), one can follow basic lines in (Wang, 2020) to identify two constants $\mathcal{K} := (\frac{2\rho L}{1-e^{-2L}})^{1/2}e^{\frac{4}{\rho}} \vee e^{2L+\frac{2}{\rho}}$, $\eta := -\frac{2}{\rho}$, only depending on $L, \rho$, but independent of $d, t$.

## 2.2. Main Results for Langevin Monte Carlo

Now we turn to the LMC and report its non-asymptotic error bound in $\mathcal{W}_2$-distance without log-concavity.

**Assumption 2.6** (Gradient Lipschitz condition). *There exists a dimension-independent constant $L_1 > 0$ such that*

$$|\nabla U(x) - \nabla U(y)| \leq L_1|x - y|, \quad \forall x, y \in \mathbb{R}^d. \tag{9}$$

Owing to (9), one can apply the triangle inequality to show that for all $x \in \mathbb{R}^d$,

$$|\nabla U(x)| \leq L_1'd^{1/2} + L_1|x|, \tag{10}$$

with $L_1'd^{1/2} := |\nabla U(0)|$. Moreover, using (9) and the Cauchy-Schwarz inequality ensures that Assumption 2.2 holds with $L = L_1$.

**Assumption 2.7.** *There exists a positive constant $\sigma_1$, independent of $d$, such that*

$$\mathbb{E}\big[|x_0|^2\big] \leq \sigma_1 d. \tag{11}$$

In addition, we need a linear growth condition of the 3rd-order derivative of $U$, which has been also used in (Li et al., 2022) in a strongly convex setting. As verified in Appendix G, a Gaussian mixture meets such a condition and the double-well potential meets a similar condition in Assumption 2.14. More examples can be found in (Li et al., 2022).

**Assumption 2.8** (Linear growth condition of the 3rd-order derivative). *There exist two positive constants $L_0'$ and $L_0$ such that for all $x, y \in \mathbb{R}^d$,*

$$|\nabla(\Delta U(x))| \leq L_0'd^{1/2} + L_0|x|. \tag{12}$$

As remarked by (Li et al., 2022), Assumption 2.8 is not necessarily stronger than Hessian Lipschitzness. By Assumption 2.6, we have

$$|\nabla^2 U(x)y| \leq L_1|y|, \quad \forall x, y \in \mathbb{R}^d. \tag{13}$$

**Lemma 2.9** (Uniform-in-time moment estimate for LMC). *Let Assumptions 2.1, 2.6 hold. If the uniform timestep satisfies $h \leq \frac{\mu}{4L_1^2} \wedge \frac{1}{\mu} \wedge 1$, then the LMC (2) satisfies*

$$\sup_{n\in\mathbb{N}_0} \mathbb{E}[|\bar{Y}_n|^2] \leq e^{-\mu t_n}\mathbb{E}[|x_0|^2] + \frac{4+4L_1'^2+2\mu'}{\mu}d. \tag{14}$$

Its proof is put in Appendix A. The first main result of this paper is as follows.

**Theorem 2.10** (Main result for LMC). *Let $\{\bar{p}_n\}_{n\in\mathbb{N}_0}$ be the semigroups associated to LMC (2) and let Assumptions 2.1, 2.3, 2.6–2.8 hold. If the uniform timestep satisfies $h \leq \frac{1}{2L} \wedge \frac{\mu}{4L_1^2} \wedge \frac{1}{\mu} \wedge 1$, then for any $n \in \mathbb{N}_0$ and initial distribution $\nu = \mathcal{L}(x_0)$, it holds*

$$\mathcal{W}_2(\nu\bar{p}_n, \pi) \leq \bar{C}_1\sqrt{d}h + \bar{C}_2\sqrt{d}e^{-\lambda nh}, \tag{15}$$

*where $\lambda := \frac{\eta}{\log\mathcal{K}+1+\eta/(2L)}$ and*

$$\begin{aligned}\bar{C}_1 &:= C(\mu, \mu', \eta, c, \sigma_1, \eta, L_0, L_0', L_1, L_1', \mathcal{K}),\\ \bar{C}_2 &:= C(\mu, \mu', c, \sigma_1, L_1'),\end{aligned} \tag{16}$$

*are explicitly given by (148).*

**Proposition 2.11.** *Let assumptions of Theorem 2.10 hold. To achieve a given accuracy tolerance $\epsilon > 0$ under $\mathcal{W}_2$-distance, a required number of iterations of the LMC (2) is of order $\widetilde{O}\big(\frac{\sqrt{d}}{\epsilon}\big)$.*

*Table 1.* Comparison of mixing times in $\mathcal{W}_2$-distance.

| | M-T | S-C | A-A |
|---|---|---|---|
| PAPERS[A] | $\widetilde{O}(d\epsilon^{-2})$ | YES | G-L |
| PAPER[B] | $\widetilde{O}(d\epsilon^{-1})$ | YES | G-L, H-L |
| PAPER[C] | $\widetilde{O}(d^{1/2}\epsilon^{-1})$ | YES | G-L, 3-RD-L-G |
| PAPER[D] | $\widetilde{O}(d\epsilon^{-4})$ | No | G-L |
| PAPER[E] | $\widetilde{O}(d\epsilon^{-1})$ | No | G-L, H-L |
| THIS WORK | $\widetilde{O}(d^{1/2}\epsilon^{-1})$ | No | G-L, 3-RD-L-G |

[A] (CHENG & BARTLETT, 2018; DALALYAN, 2017A; DURMUS ET AL., 2019).
[B] (DURMUS & MOULINES, 2019).
[C] (LI ET AL., 2022).
[D] (MAJKA ET AL., 2020).
[E] (MOU ET AL., 2022).
M-T: MIXING TIME. S-C: STRONG CONVEXITY. A-A: ADDITIONAL ASSUMPTION. G-L: GRADIENT LIPSCHITZNESS. H-L: HESSIAN LIPSCHITZNESS. 3-RD-L-G: LINEAR GROWTH OF THE 3-RD DERIVATIVE.

See Appendix E for proofs of Theorem 2.10 and Proposition 2.11. In Table 1, we compare the number of iterations of LMC algorithm (2) required to achieve $\epsilon$ error in $\mathcal{W}_2$-distance in the literature. Clearly, our error bounds match the best ones in the strongly log-concave case and improve upon the best-known convergence rates in non-convex settings.

### 2.3. Main Results for Modified Langevin Monte Carlo

Next we consider the case when the gradient of the potential $U$ is non-globally Lipschitz with superlinear growth.

**Assumption 2.12** (Gradient polynomial growth condition). There exists a positive constant $\mathbb{L}_1$, independent of $d$, such that for all $x, y \in \mathbb{R}^d$, $\gamma > 0$,

$$|\nabla U(x) - \nabla U(y)| \leq \mathbb{L}_1(1 + |x|^{\gamma} + |y|^{\gamma})|x - y|. \quad (17)$$

Thanks to (17), the triangle inequality and the Young inequality, we infer

$$|\nabla U(x)| \leq \mathbb{L}_1' d^{1/2} + 2\mathbb{L}_1|x|^{\gamma+1}, \quad \forall x \in \mathbb{R}^d, \quad (18)$$

where $\mathbb{L}_1' d^{1/2} := |\nabla U(0)| + \gamma\mathbb{L}_1$. Similar to (13), we get

$$|\nabla^2 U(x)y| \leq \mathbb{L}_1(1 + |x|^{\gamma})|y|. \quad (19)$$

**Assumption 2.13.** There exists some positive constant $\sigma_2(p)$, independent of $d$, such that

$$\mathbb{E}\big[|x_0|^{2p}\big] \leq \sigma_2(p)d^p, \quad p \in [1, \tfrac{11\gamma+10}{2}], \quad (20)$$

where $\gamma$ comes from Assumption 2.12.

**Assumption 2.14** (Polynomial growth condition of the 3rd-order derivative). There exist two positive constants $\mathbb{L}_0'$ and $\mathbb{L}_0$, independent of $d$, such that

$$|\nabla(\Delta U(x))| \leq \mathbb{L}_0' d^{(\gamma+1)/2} + \mathbb{L}_0 d^{\gamma_1/2}|x|^{\gamma_2} \quad (21)$$

holds for all $x, y \in \mathbb{R}^d$, where $\gamma_1 \geq 0, \gamma_2 \geq 1$ obeying $\gamma_1 + \gamma_2 = \gamma + 1$.

For this case, we consider a kind of modified Langevin Monte Carlo, also termed as projected Langevin Monte Carlo (pLMC), introduced by (Pang et al., 2025) and given by, $\check{Y}_0 = x_0$,

$$\check{Y}_{n+1} = \mathcal{T}^h(\check{Y}_n) - \nabla U(\mathcal{T}^h(\check{Y}_n))h + \sqrt{2h}\zeta_{n+1}, \quad (22)$$

where $\zeta_k := (\zeta_k^1, \zeta_k^2, \cdots, \zeta_k^d)^T$ are i.i.d standard d-dimensional Gaussian vectors, $\mathcal{T}^h : \mathbb{R}^d \to \mathbb{R}^d$ is a projection operator, defined by, for a dimension-independent parameter $\vartheta \geq 1$ and any $x \in \mathbb{R}^d$,

$$\mathcal{T}^h(x) := \begin{cases} \min\left\{1, \vartheta d^{\frac{1}{2(\gamma+1)}} h^{-\frac{1}{2(\gamma+1)}} |x|^{-1}\right\} x, & x \neq 0, \\ 0, & x = 0, \end{cases} \quad (23)$$

with $\gamma$ being given in Assumption 2.12.

**Lemma 2.15** (Uniform-in-time moment estimate for pLMC). *Let Assumptions 2.1, 2.12, hold. For the uniform timestep satisfying $h \leq \frac{1}{2\mu} \wedge \frac{2\mu}{\mu+2(\mathbb{L}_1'+2\mathbb{L}_1)} \wedge 1$, the pLMC (22) has uniform-in-time moment bounds, for any $p \in [0, \infty) \cap \mathbb{N}$,*

$$\sup_{n \in \mathbb{N}_0} \mathbb{E}[|\check{Y}_n|^{2p}] \leq e^{-\frac{\mu}{2}t_n}\mathbb{E}[|x_0|^{2p}] + C_{\check{1}}d^p, \quad (24)$$

*where $t_n := nh$, $C_{\check{1}} := C(\mu, \mu', \gamma, \vartheta, \mathbb{L}_1, \mathbb{L}_1', p)$.*

Its proof can be found in Lemma 3.4 of (Pang et al., 2025).

**Theorem 2.16** (Main result for pLMC). *Let $\{\check{p}_n\}_{n \in \mathbb{N}_0}$ be the semigroups associated to pLMC (22) and Assumptions 2.1–2.3, 2.12–2.14 hold. If the uniform timestep satisfies $h \leq \frac{1}{2L} \wedge \frac{1}{\mu} \wedge \frac{2\mu}{\mu+2(\mathbb{L}_1'+2\mathbb{L}_1)^2} \wedge 1$, then for any $n \in \mathbb{N}_0$ and initial distribution $\nu = \mathcal{L}(x_0)$, it holds*

$$\mathcal{W}_2(\nu\check{p}_n, \pi) \leq \check{C}_1 d^{(11\gamma+2)/4}h + \check{C}_2 d^{1/2}e^{-\lambda nh}, \quad (25)$$

*where $\lambda := \frac{\eta}{\log\mathcal{K}+1+\eta/(2L)}$ and*

$$\begin{aligned} \check{C}_1 &:= C(\mu, \mu', \eta, c, \gamma, \sigma_2, \vartheta, L, \mathbb{L}_0, \mathbb{L}_0', \mathbb{L}_1, \mathbb{L}_1', \mathcal{K}), \\ \check{C}_2 &:= C(\mu, \mu', c, \gamma, \sigma_2, \vartheta, \mathbb{L}_1, \mathbb{L}_1'). \end{aligned} \quad (26)$$

See Appendix F for the proof of this theorem. This theorem also help us to obtain the mixing time, whose proof is similar to Proposition 2.11 and thus omitted here.

**Proposition 2.17.** *Let assumptions of Theorem 2.16 hold. To achieve a given accuracy tolerance $\epsilon > 0$ under $\mathcal{W}_2$-distance, a required number of iterations of the pLMC (22) is of order $\widetilde{O}\big(\frac{d^{(11\gamma+2)/4}}{\epsilon}\big)$.*

## 2.4. Overview of Proof

In this subsection we present an overview of the above non-asymptotic error bounds. For an approximation $\{\tilde{Y}_n\}_{n\in\mathbb{N}_0}$ to the SDE $\{X_t\}_{t\geq 0}$, the goal of long-time error analysis is to bound $\mathcal{W}_2(\nu\tilde{p}_n, \pi)$, where $\pi \in \mathcal{P}(\mathbb{R}^d)$ is the invariant distribution of $\{p_t\}_{t\geq 0}$ and $\{\tilde{p}_n\}_{n\in\mathbb{N}_0}$ is the transition semigroups associated to $\{\tilde{Y}_n\}_{n\in\mathbb{N}_0}$. By the triangle inequality, we have

$$
\begin{aligned}
\mathcal{W}_2(\nu\tilde{p}_n, \pi) \leq &\mathcal{W}_2(\nu\tilde{p}_{n-n_1}\tilde{p}_{n_1}, \nu\tilde{p}_{n-n_1}p_{t_{n_1}}) \\
&+ \mathcal{W}_2(\nu\tilde{p}_{n-n_1}p_{t_{n_1}}, \pi), \ n \geq n_1.
\end{aligned} \tag{27}
$$

Following the triangle inequality, we give an overview of five steps that comprise the proof of Theorems 2.10, 2.16.

**Step 1.** As proved in Lemma 3.1, the solutions of SDEs have uniform-in-time moment estimates with the help of dissipativity conditions. In addition, we require the numerical approximations to be uniform-in-time moment bounded (cf. Condition (A2)). See Subsection 3.2 for details.

**Step 2.** We establish the finite-time mean-square fundamental convergence theorem for general SDEs (see Theorem 3.3), which is used to deal with the first term on the right-hand side of (27):

$$
\mathcal{W}_2(\nu\tilde{p}_{n-n_1}\tilde{p}_{n_1}, \nu\tilde{p}_{n-n_1}p_{t_{n_1}}) \leq C(t_{n_1})h^{p_2-\frac{1}{2}}, \tag{28}
$$

where we explicitly show the dependence of the error constant $C(t_{n_1})$ on $t_{n_1}$ (cf. (48)). See Subsection 3.3 for more details.

**Step 3.** To estimate the second term on the right-hand side of (27), we rely on the exponential ergodicity of the SDE (see Condition (A5)) to obtain:

$$
\mathcal{W}_2(\nu\tilde{p}_{n-n_1}p_{t_{n_1}}, \pi) \leq \mathcal{K}e^{-\eta t_{n_1}}\mathcal{W}_2(\nu\tilde{p}_{n-n_1}, \pi). \tag{29}
$$

See Subsection 3.4 for details.

**Step 4.** Collecting (28) and (29) together and choosing $t_{n_1} = \chi_0$ such that $\mathcal{K}e^{-\eta\chi_0} = \frac{1}{e}$, one can derive from the uniform-in-time bounded moments that

$$
\mathcal{W}_2(\nu\tilde{p}_n, \pi) \leq C(\chi_0)h^{p_2-\frac{1}{2}} + \frac{1}{e}\mathcal{W}_2(\nu\tilde{p}_{n-n_1}, \pi). \tag{30}
$$

By iteration and using Lemma D.1, one can deduce

$$
\mathcal{W}_2(\nu\tilde{p}_n, \pi) \leq \hat{\mathcal{K}}_1 h^{p_2-\frac{1}{2}} + \hat{\mathcal{K}}_2 e^{-\lambda_0 nh}. \tag{31}
$$

See Subsection 3.5 for details.

**Step 5.** We verify all conditions required by the non-convex theoretical framework of uniform-in-time convergence theorem in Section 3 and calculate all constants to obtain the expected non-asymptotic convergence rate of the underlying LMC (2) and pLMC (22) for Langevin SDE. See Appendix E and F for details.

# 3. A Non-Convex Theoretical Framework of Uniform-in-Time Convergence Theorem

The aim of the present section is to construct a non-convex theoretical framework of uniform-in-time convergence theorem for general SDEs, which will help us to easily analyze the non-asympiotic error bounds of schemes in Section 2. To this end, we set up a general framework by introducing general SDEs and their numerical approximation as follows.

## 3.1. SDEs and Their Numerical Approximations

We consider the following Itô SDEs as follows:

$$
\mathrm{d}X_t = f(X_t)\,\mathrm{d}t + \sum_{k=1}^m g^k(X_t)\,\mathrm{d}W_t^k, \ X_0 = x_0', \tag{32}
$$

where $t \geq 0$, $f\colon \mathbb{R}^d \to \mathbb{R}^d$ is a drift function, $g = (g^1, g^2, \cdots, g^m)\colon \mathbb{R}^d \to \mathbb{R}^{d\times m}$ is a diffusion function and $\{W_t = (W_t^1, W_t^2, \cdots, W_t^m)\}_{t\geq 0}$ is a $m$-dimensional Wiener process. Let $X_t := X(s, x; t) = X_{s,x}(t), 0 \leq s \leq t$, denote the solution to (32) at $t$, starting from the initial value $x$ at $s$, given by

$$
X_{s,x}(t) = x + \int_s^t f(X_r)\,\mathrm{d}r + \sum_{k=1}^m \int_s^t g^k(X_r)\,\mathrm{d}W_r^k. \tag{33}
$$

To approximate SDE (32), we introduce the one-step approximation $Y(t, x; t+h)$ for the solution $X(t, x; t+h)$ of SDE (32) in the form of, for all $x \in \mathbb{R}^d$

$$
Y(t, x; t+h) = x + \Phi(t, x, h; \xi_t), \tag{34}
$$

where $h$ is uniform timestep and $\xi_t$ is a random vector defined on $(\Omega, \mathcal{F}, \mathbb{P})$ with moments of a sufficiently high order and $\Phi$ is a function from $[0, +\infty) \times \mathbb{R}^d \times (0, +\infty) \times \mathbb{R}^m$ to $\mathbb{R}^d$. Using the one-step approximation (34), we recurrently construct numerical approximations $\{Y_n\}_{n\in\mathbb{N}_0}$ on the uniform mesh grid $\{t_n = nh\}_{n\in\mathbb{N}_0}$, given by

$$
Y_0 = x_0', \quad Y_{n+1} = Y_n + \Phi(t_n, Y_n, h; \xi_n), \tag{35}
$$

where $\xi_n$ is independent of $Y_0, Y_1, \ldots, Y_n, \xi_0, \xi_1, \ldots, \xi_{n-1}$ for all $n \geq 1$.

## 3.2. Uniform-in-Time Bounded Moments

Under the following dissipative condition, the general SDE (32) have uniform-in-time bounded moments.

**Condition (A1)** There exist some constants $p^* \in [1, \infty)$, $\hat{\mu}^*, \mu^* \in (0, \infty)$ such that drift and diffusion coefficients of the SDE (32) satisfy, for all $x \in \mathbb{R}^d$,

$$
\langle x, f(x) \rangle + \frac{2p^*-1}{2}\|g(x)\|_F^2 \leq \hat{\mu}^* - \mu^*|x|^2. \tag{36}
$$

Moreover, the initial value obeys $\mathbb{E}[|x_0'|^{2p^*}] < \infty$.

We highlight that constants used here (i.e., $\hat{\mu}^*, \mu^*$) and throughout this section might depend on the dimensions $d, m$. When all established results in this section are applied to the Langevin SDE and its numerical approximation, the dependence will be explicitly given. Thanks to this condition, one can easily derive the uniform-in-time bounded moments for SDEs (32) whose proof is put in Appendix B.

**Lemma 3.1.** *Let Condition (A1) hold and $\{X_t\}_{t\geq 0}$ denote the solution of SDE* (32). *Then for any $p \in [1, p^*]$, it holds*

$$\sup_{t\geq 0} \mathbb{E}\big[|X_t|^{2p}\big] \leq C_0^* \mathbb{E}\big[|x_0'|^{2p}\big] + \hat{C}_0^*(p), \qquad (37)$$

*where $c^* \in (0, 2\mu^*)$, $C_0^* \geq 1$ and*

$$\hat{C}_0^*(p) := \tfrac{2}{c^* p}\big(\tfrac{2p-2}{(2\mu^*-c^*)p}\big)^{p-1}\hat{\mu}^{*p}. \qquad (38)$$

Moreover, we need the following assumption on bounded moments of numerical solutions.

**Condition (A2)** There exist some positive constants $h_0$, $C_1^* \geq 1$ and $\hat{C}_1^*(p)$ such that the numerical solution (35) has uniform-in-time moments, i.e., for any $p \geq 1$, $h \leq h_0$,

$$\sup_{n\in\mathbb{N}_0} \mathbb{E}\big[|Y_n|^{2p}\big] \leq C_1^* \mathbb{E}\big[|x_0'|^{2p}\big] + \hat{C}_1^*(p). \qquad (39)$$

### 3.3. The Finite-Time Mean-Square Fundamental Convergence Theorem for SDEs

This section revisits the finite-time mean-square fundamental convergence theorem. Following the idea of the strong convergence theorem originally proposed by Milstein (Milstein, 1988), we aim to reformulate a general mean-square convergence theorem for one-step approximations of SDEs (32). First, we assume the monotonicity and polynomial growth conditions as follows.

**Condition (A3)** There exists a constant $L^* > 0$ such that drift and diffusion coefficients of SDEs (32) satisfy, $\forall x, y \in \mathbb{R}^d$,

$$\langle x-y, f(x)-f(y)\rangle + \tfrac{1}{2}\|g(x)-g(y)\|_F^2 \leq L^*|x-y|^2. \quad (40)$$

**Condition (A4)** There exists a positive constant $L_f^*$ such that the drift coefficients satisfy, for all $x, y \in \mathbb{R}^d$, $r_0 \in [0, \infty)$,

$$|f(x) - f(y)| \leq L_f^*\big(1 + |x|^{r_0} + |y|^{r_0}\big)|x - y|. \quad (41)$$

Condition (A4) immediately implies that

$$|f(x)| \leq \hat{L}_f^* + 2L_f^*|x|^{r_0+1}, \quad \forall x \in \mathbb{R}^d \qquad (42)$$

where $\hat{L}_f^* = |f(0)| + r_0 L_f^*$. Based on these two conditions, one can easily obtain the next lemma, whose proof is put in Appendix B.

**Lemma 3.2.** *Let Conditions (A1), (A3), (A4) hold and let $X_{t,x}(t+\theta)$, $X_{t,y}(t+\theta)$ be two solutions of SDEs* (32) *starting from $x, y$. Let the uniform timestep $h > 0$ satisfy $2L^*h \leq 1$. Then for any $t \geq 0$, $\theta > 0$, the following representations*

$$\begin{aligned}S_{t,x,y}(t+\theta) :=& X_{t,x}(t+\theta) - X_{t,y}(t+\theta),\\ Z_{t,x,y}(t+\theta) :=& S_{t,x,y}(t+\theta) - (x-y),\end{aligned} \qquad (43)$$

*satisfy*

$$\mathbb{E}\big[|S_{t,x,y}(t+h)|^2\big] \leq (1+4L^*h)|x-y|^2, \qquad (44)$$

$$\begin{aligned}\mathbb{E}\big[|Z_{t,x,y}(t+h)|^2\big] \leq & \big(6L^* + 6L_f^*(\hat{C}_{r_0}^* + C_0^*|x|^{2r_0}\\ & + C_0^*|y|^{2r_0})^{1/2}\big)|x-y|^2 h,\end{aligned} \qquad (45)$$

*where $\hat{C}_{r_0}^*$ is given by* (79).

Now, we derive the following finite-time strong convergence theorem.

**Theorem 3.3.** *Let Conditions (A1)–(A4) hold and let the uniform timestep $h > 0$ satisfy $h \leq \frac{1}{2L^*} \wedge h_0 \wedge 1$. For any fixed $n_1 \in \mathbb{N}$, we let $T := n_1 h$. Let $\{X_t\}_{t\in[0,T]}$ and $\{Y_n\}_{n\in[n_1]}$ denote solutions of SDEs* (32) *and numerical approximations* (35), *respectively. Suppose that the one-step approximation* (34) *has local weak and strong error of order $p_1$ and $p_2$, respectively, with $p_2 \geq \frac{1}{2}$, $p_1 \geq p_2 + \frac{1}{2}$, i.e., there exist some positive constants $\hat{K}_1^*, K_1^*, \hat{K}_2^*, K_2^* > 0$ and $r \geq 1$ such that for all $0 \leq t \leq T - h$, $x \in \mathbb{R}^d$,*

$$\begin{aligned}&\Big|\mathbb{E}\big[X(t,x;t+h) - Y(t,x;t+h)\big]\Big|\\ \leq & \big(\hat{K}_1^* + K_1^*|x|^{2r}\big)^{1/2} h^{p_1},\\ &\Big(\mathbb{E}\big[|X(t,x;t+h) - Y(t,x;t+h)|^2\big]\Big)^{1/2}\\ \leq & \big(\hat{K}_2^* + K_2^*|x|^{2r}\big)^{1/2} h^{p_2}.\end{aligned} \qquad (46)$$

*Then*

$$\begin{aligned}&\sup_{n\in[n_1]} \Big(\mathbb{E}\big[|X(0,x_0';t_n) - Y(0,x_0';t_n)|^2\big]\Big)^{1/2}\\ \leq & C^*(T)\Big(\hat{K}^* + K^*\mathbb{E}\big[|x_0'|^{2r+r_0}\big]\Big)^{1/2} h^{p_2-1/2},\end{aligned} \qquad (47)$$

*where*

$$C^*(T) := e^{\frac{1}{2}(1+10L^*+6L_f^*)T}, \qquad (48)$$

*and $\hat{K}^*$ and $K^*$ are explicitly given by* (103).

This theorem is proved in Appendix C. Similar results were obtained by (Tretyakov & Zhang, 2013), where the parameter (e.g., $T$) dependence in the error constants was not explicitly shown. We mention that both finite-time convergence theorems follow the original idea of (Milstein, 1988).

## 3.4. Exponential Ergodicity in $\mathcal{W}_2$-Distance

In the present subsection, we assume that the general SDEs (32) have exponential ergodicity. Let $\{P_t\}_{t\geq 0}$ be the Markov semigroup associated to the solutions $\{X_t\}_{t\geq 0}$ of SDE (32), which is defined by, for all $x \in \mathbb{R}^d$ and $0 \leq s \leq t$

$$P_t\phi(x) := \mathbb{E}\big[\phi(X_{s,x}(t))\big], \quad \phi \in B_b(\mathbb{R}^d). \quad (49)$$

We put some conditions on exponential ergodicity in $\mathcal{W}_2$-distance of SDEs.

**Condition (A5)** Assume the semigroup $\{P_t\}_{t\geq 0}$ associated to SDE (32) has a unique invariant distribution $\Pi$. There exists two positive constants $\mathcal{K}^*$ and $\eta^*$ such that, for all $t \geq 0$,

$$\mathcal{W}_2(\nu'P_t, \Pi) \leq \mathcal{K}^* e^{-\eta^* t} \mathcal{W}_2(\nu', \Pi), \quad (50)$$

where $\nu' := \mathcal{L}(x_0')$ is the initial distribution.

In (Wang, 2020), some sufficient conditions are provided on coefficients of SDEs such that the underlying SDEs have exponential ergodicity in $\mathcal{W}_2$-distance. In particular, the Langevin SDE (1) has such an exponential ergodicity property in our non-convex setting.

## 3.5. The Uniform-in-Time Convergence Theorem

Under the above conditions, we are able to formulate the uniform-in-time convergence theorem as follows.

**Theorem 3.4.** *Let conditions of Theorem 3.3 and Condition (A5) hold. Let $\{\tilde{P}_n\}_{n \in \mathbb{N}_0}$ be the semigroup associated to the numerical scheme (35). Then for any $n \in \mathbb{N}_0$ and any initial distribution $\nu' = \mathcal{L}(x_0')$, it holds*

$$\mathcal{W}_2(\nu'\tilde{P}_n, \Pi) \leq \hat{\mathcal{K}}_1^* h^{p_2 - \frac{1}{2}} + \hat{\mathcal{K}}_2^* e^{-\lambda^* nh}, \quad (51)$$

*where $\lambda^* := \frac{\eta^*}{\log \mathcal{K}^* + 1 + \eta^*/(2L^*)}$ and the constants $\hat{\mathcal{K}}_1^*$, $\hat{\mathcal{K}}_2^*$ are given by (122).*

The proof is postponed to Appendix D.

# 4. Numerical Experiments

In this section, numerical results are performed to verify the above theoretical finding. To this end, we consider two target measures specified by the following two potentials:

a). Gaussian Mixture,

$$U_1(x) = \tfrac{1}{2}|x - a|^2 - \log\big(1 + e^{-2\langle x,a\rangle}\big), \quad x \in \mathbb{R}^d, \quad (52)$$

for a given $a \in \mathbb{R}^d$.

b). Double-well potential,

$$U_2(x) = \tfrac{\alpha}{4}|x|^4 - \tfrac{\beta}{2}|x|^2, \quad x \in \mathbb{R}^d. \quad (53)$$

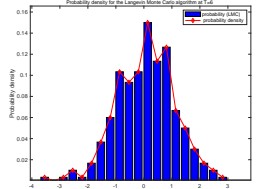

(a) Probability density of the first component for the Gaussian Mixture.

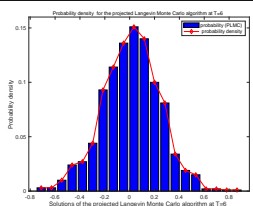

(b) Probability density of the first component for the Double-well potential.

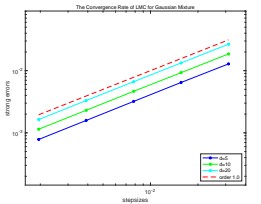

(c) Strong convergence rates of LMC algorithm for the Gaussian Mixture.

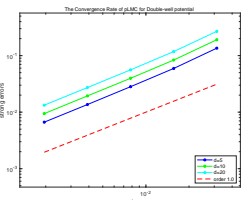

(d) Strong convergence rates of pLMC algorithm for the Double-well potential.

We can directly show that the Gaussian mixture satisfies Assumption 2.1 with $\mu = \tfrac{1}{2}$, $\mu'd = 2|a|^2$, Assumption 2.8 with $L_0' = 0$, $L_0 = \tfrac{8|a|^4}{3}$, Assumption 2.6 with $L_1 = 1 + 4|a|^2$ and the double-well potential with $\alpha = \beta = 1$ satisfies the Assumption 2.1 with $\mu = 1$, $\mu'd = 1$, Assumption 2.2 with $L = 4\sqrt{2} + 19/2$, Assumption 2.12 with $\gamma = 2$ and $\mathbb{L}_1 = 1$, Assumption 2.14 with $\mathbb{L}_0' = 0$, $\mathbb{L}_0 = 4$, $\gamma_1 = \gamma_2 = 1$. For these two cases, strongly convex outside a ball is satisfied and thus the log-Sobolev inequality is satisfied. Assumptions 2.1, 2.2, 2.6, 2.12 have been already verified in Proposition 3.1 of (Neufeld et al., 2025) and Example 2.5 in (Pang et al., 2025). In Appendix G, we give a brief proof of Assumption 2.8 for Gaussian mixture and Assumption 2.14 for the double-well potential.

In what follows we set $x_0 = 0$, $a \in \mathbb{R}^d$ with $|a| = 2$ and $\alpha = 1$, $\beta = 16$, where all components of the $a$ are equal. We emphasize that the potential $U_1$ of the Gaussian mixture is non-convex in our setting (see Example 1 of (Dalalyan, 2017b) for a convex setting, i.e., $|a| < 1$).

To study the probability distributions, we fix the dimension $d = 100$, $T = 6$ and simulate 300 independent Markov chains using LMC algorithm for each step size $h = \{0.001, 0.005, 0.01, 0.05, 0.1\}$. In Figure 1(a) and 1(b), we present the density curves of the first components for Gaussian mixture and double-well potential, respectively.

Moving on to the convergence analysis, we run the LMC algorithm (2) and pLMC algorithm (22) for the Langevin SDE by different stepsizes $h$ till $T = 6$. Here numerical approximations are performed using five different stepsizes $h \in \{2^{-5}, 2^{-6}, 2^{-7}, 2^{-8}, 2^{-9}\}$. The *exact* solution is identified as the numerical one using a fine stepsize $h_{ref} = 2^{-13}$ and the expectations are approximated by computing aver-

ages over 3000 samples. From Figures 1(c), 1(d), it is observed that the convergence rate is of order 1.

## 5. Conclusion and Future Work

In the present article, we provide an error bound $O(\sqrt{d}h)$ in $\mathcal{W}_2$-distance for the classical LMC without log-concavity. For the case when the gradient of the potential $U$ is non-globally Lipschitz with superlinear growth, modified LMC samplers are introduced and analyzed, with a $\mathcal{W}_2$ error bound obtained. These results are derived essentially based on a newly developed non-convex theoretical framework of uniform-in-time convergence for discretizations of general SDEs. This framework can be also applied to other metrics and higher order LMC sampling algorithms (Sabanis & Zhang, 2019; Li et al., 2019), which is our ongoing work.

## Acknowledgements

The authors thank Lei Dai, Yingsong Jiang, Chenxu Pang, Xiaoyan Zhang, Yuying Zhao, anonymous reviewers and area chair for suggestions that significantly improved the quality of this paper. This work was supported by Natural Science Foundation of China (Nos. 12471394, 12071488, 12371417).

## Impact Statement

This paper presents work whose goal is to advance the field of Machine Learning. There are many potential societal consequences of our work, none which we feel must be specifically highlighted here.

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

## A. Proofs of Lemmas in Section 2

*Proof of Lemma 2.4.* Under Assumption 2.1, one can easily check that, for any $p^* \geq 1$, Condition (**A1**) is fulfilled with

$$\hat{\mu}^* = \mu'd + (2p^* - 1)d, \quad \mu^* = \mu. \tag{54}$$

Then Lemma 2.4 is a direct consequence of Lemma 3.1. $\square$

*Proof of Lemma 2.9.* Taking square on both sides of the LMC (2), one can use Assumption (2.1), (10) and the Cauchy-Schwarz inequality to derive that, for $h \leq \frac{\mu}{4L_1^2} \wedge 1$,

$$
\begin{aligned}
|\bar{Y}_{n+1}|^2 =& |\bar{Y}_n|^2 + h^2|\nabla U(\bar{Y}_n)|^2 + 2h|\zeta_{n+1}|^2 - 2h\langle \bar{Y}_n, \nabla U(\bar{Y}_n)\rangle + 2\sqrt{2h}\langle \bar{Y}_n, \zeta_{n+1}\rangle - 2\sqrt{2h}h\langle \nabla U(\bar{Y}_n), \zeta_{n+1}\rangle \\
\leq& |\bar{Y}_n|^2 + 2h^2|\nabla U(\bar{Y}_n)|^2 + 4h|\zeta_{n+1}|^2 - 2h\langle \bar{Y}_n, \nabla U(\bar{Y}_n)\rangle + 2\sqrt{2h}\langle \bar{Y}_n, \zeta_{n+1}\rangle \\
\leq& |\bar{Y}_n|^2 + 4L_1^2 h^2|\bar{Y}_n|^2 + 4h|\zeta_{n+1}|^2 - 2\mu h|\bar{Y}_n|^2 + 2\sqrt{2h}\langle \bar{Y}_n, \zeta_{n+1}\rangle + (4L_1'^2 + 2\mu')dh \\
\leq& (1 - \mu h)|\bar{Y}_n|^2 + 4h|\zeta_{n+1}|^2 + 2\sqrt{2h}\langle \bar{Y}_n, \zeta_{n+1}\rangle + (4L_1'^2 + 2\mu')dh.
\end{aligned}
\tag{55}
$$

Taking expectations on both sides of (55) we deduce

$$
\begin{aligned}
\mathbb{E}\big[|\bar{Y}_{n+1}|^2\big] \leq& (1 - \mu h)\mathbb{E}\big[|\bar{Y}_n|^2\big] + 4h\mathbb{E}\big[|\zeta_{n+1}|^2\big] + (4L_1'^2 + 2\mu')dh \\
\leq& (1 - \mu h)\mathbb{E}\big[|\bar{Y}_n|^2\big] + (4 + 4L_1'^2 + 2\mu')dh.
\end{aligned}
\tag{56}
$$

By iteration and by the assumption $h \leq \frac{1}{\mu}$, one can use the inequality $1 - x \leq e^{-x}, 0 \leq x \leq 1$ to arrive at

$$
\begin{aligned}
\mathbb{E}\big[|\bar{Y}_{n+1}|^2\big] \leq& (1 - \mu h)^{n+1}\mathbb{E}\big[|x_0|^2\big] + (4 + 4L_1'^2 + 2\mu')dh \sum_{i=0}^{n}(1 - \mu h)^i \\
\leq& e^{-\mu t_{n+1}}\mathbb{E}[|x_0|^2] + \frac{4 + 4L_1'^2 + 2\mu'}{\mu}d.
\end{aligned}
\tag{57}
$$

The proof is now completed. $\square$

## B. Proofs of Lemmas in Section 3

*Proof of Lemma 3.1.* For every integer $n \geq 1$, we first define a stopping time as follows:

$$\tau_n := \inf\big\{s \geq 0 : |X_s| \geq n\big\}. \tag{58}$$

By using the Itô formula and the Cauchy-Schwarz inequality, we can derive that, for any $t \geq 0$, $\varepsilon_1 > 0$ and $c^* \in (0, 2\mu^*)$,

$$
\begin{aligned}
& e^{c^* p(t \wedge \tau_n)}\big(\varepsilon_1 + |X_{t \wedge \tau_n}|^2\big)^p \\
=& \big(\varepsilon_1 + |x_0'|^2\big)^p + c^* p \int_0^{t \wedge \tau_n} e^{c^* ps}\big(\varepsilon_1 + |X_s|^2\big)^p ds + 2p \int_0^{t \wedge \tau_n} e^{c^* ps}\big(\varepsilon_1 + |X_s|^2\big)^{p-1}\langle X_s, f(X_s)\rangle ds \\
& + p \int_0^{t \wedge \tau_n} e^{c^* ps}\big(\varepsilon_1 + |X_s|^2\big)^{p-1}\|g(X_s)\|_F^2 ds + 2p(p-1) \int_0^{t \wedge \tau_n} e^{c^* ps}\big(\varepsilon_1 + |X_s|^2\big)^{p-2} \sum_{k=1}^{m} \langle X_s, g^k(X_s)\rangle^2 ds \\
& + 2p \int_0^{t \wedge \tau_n} e^{c^* ps}\big(\varepsilon_1 + |X_s|^2\big)^{p-1} \sum_{k=1}^{m} \langle X_s, g^k(X_s)\rangle dW_s^k \\
\leq& \big(\varepsilon_1 + |x_0'|^2\big)^p + c^* p \int_0^{t \wedge \tau_n} e^{c^* ps}\big(\varepsilon_1 + |X_s|^2\big)^p ds + 2p \int_0^{t \wedge \tau_n} e^{c^* ps}\big(\varepsilon_1 + |X_s|^2\big)^{p-1} \sum_{k=1}^{m} \langle X_s, g^k(X_s)\rangle dW_s^k \\
& + p \int_0^{t \wedge \tau_n} e^{c^* ps}\big(\varepsilon_1 + |X_s|^2\big)^{p-1}\big(2\langle X_s, f(X_s)\rangle + (2p-1)\|g(X_s)\|_F^2\big) ds.
\end{aligned}
\tag{59}
$$

Taking expectations on both sides and letting $\varepsilon_1 \to 0$ yield

$$
\begin{aligned}
\mathbb{E}\big[e^{c^* p(t \wedge \tau_n)}|X_{t \wedge \tau_n}|^{2p}\big] \leq& \mathbb{E}\big[|x_0'|^{2p}\big] + c^* p \mathbb{E}\bigg[\int_0^{t \wedge \tau_n} e^{c^* ps}|X_s|^{2p} ds\bigg] + p \mathbb{E}\bigg[\int_0^{t \wedge \tau_n} e^{c^* ps}|X_s|^{2p-2} \\
& \times \big(2\langle X_s, f(X_s)\rangle + (2p-1)\|g(X_s)\|_F^2\big) ds\bigg],
\end{aligned}
\tag{60}
$$

where the property of the Itô integral was used that

$$\mathbb{E}\left[\int_0^{t\wedge\tau_n} e^{c^*ps}|X_s|^{2p-2}\sum_{k=1}^m\langle X_s, g^k(X_s)\rangle\,\mathrm{d}W_s^k\right] = 0. \tag{61}$$

Taking (36) into account, one can easily derive from (60) that

$$\mathbb{E}\left[e^{c^*p(t\wedge\tau_n)}|X_{t\wedge\tau_n}|^{2p}\right] \leq \mathbb{E}\left[|x_0'|^{2p}\right] + p\mathbb{E}\left[\int_0^{t\wedge\tau_n}(c^* - 2\mu^*)e^{c^*ps}|X_s|^{2p}\mathrm{d}s\right] + 2p\mathbb{E}\left[\int_0^{t\wedge\tau_n}\hat{\mu}^*e^{c^*ps}|X_s|^{2p-2}\mathrm{d}s\right]. \tag{62}$$

It follows from the Young inequality that, for any $\varepsilon_2 > 0$

$$\hat{\mu}^*|X_s|^{2p-2} \leq \varepsilon_2|X_s|^{2p} + \tfrac{1}{p}\left(\tfrac{p-1}{\varepsilon_2 p}\right)^{p-1}\hat{\mu}^{*p}. \tag{63}$$

Setting $\varepsilon_2 := \frac{2\mu^*-c^*}{2}$, one can further insert (63) into (62) to obtain

$$\mathbb{E}\left[e^{c^*p(t\wedge\tau_n)}|X_{t\wedge\tau_n}|^{2p}\right] \leq \mathbb{E}\left[|x_0'|^{2p}\right] + \tfrac{2}{c^*p}\left(\tfrac{2p-2}{(2\mu^*-c^*)p}\right)^{p-1}\hat{\mu}^{*p}\,\mathbb{E}\left[e^{c^*p(t\wedge\tau_n)}\right]. \tag{64}$$

Thanks to the Fatou lemma, we then let $n\to\infty$ to attain

$$\mathbb{E}\left[|X_t|^{2p}\right] \leq e^{-c^*pt}\mathbb{E}\left[|x_0'|^{2p}\right] + \tfrac{2}{c^*p}\left(\tfrac{2p-2}{(2\mu_*-c^*)p}\right)^{p-1}\hat{\mu}^{*p}. \tag{65}$$

Observing $C_0^* \geq 1 \geq e^{-c^*pt}$, one can easily obtain the desired assertion. $\qquad\square$

*Proof of Lemma 3.2.* We first prove (44). By using the Itô formula and the condition (40), we obtain for any $h > 0$,

$$\begin{aligned}
&\mathbb{E}\left[|S_{t,x,y}(t+h)|^2\right]\\
=&\mathbb{E}\left[|x-y|^2\right] + 2\int_t^{t+h}\mathbb{E}\left[\langle X_{t,x}(s) - X_{t,y}(s), f(X_{t,x}(s)) - f(X_{t,y}(s))\rangle\right]\mathrm{d}s\\
&+ \int_t^{t+h}\mathbb{E}\left[\|g(X_{t,x}(s)) - g(X_{t,y}(s))\|_F^2\right]\mathrm{d}s\\
\leq&\mathbb{E}\left[|x-y|^2\right] + 2L^*\int_t^{t+h}\mathbb{E}\left[|S_{t,x,y}(s)|^2\right]\mathrm{d}s.
\end{aligned} \tag{66}$$

The Gronwall inequality helps us to show

$$\mathbb{E}\left[|S_{t,x,y}(t+h)|^2\right] \leq e^{2L^*h}|x-y|^2. \tag{67}$$

We notice that for any $0 < h \leq \frac{1}{2L^*}$,

$$e^{2L^*h} = \sum_{i=0}^{\infty}\frac{(2L^*h)^i}{i!} = 1 + 2L^*h\sum_{i=1}^{\infty}\frac{(2L^*h)^{i-1}}{i!} \leq 1 + 2L^*(e-1)h \leq 1 + 4L^*h. \tag{68}$$

Inserting this into (67) yields (44). Regarding $\mathbb{E}[|Z_{t,x,y}(t+h)|^2]$, by applying the Itô formula, we have

$$\begin{aligned}
\mathbb{E}\left[|Z_{t,x,y}(t+h)|^2\right] =&2\int_t^{t+h}\mathbb{E}\left[\langle Z_{t,x,y}(s), f(X_{t,x}(s)) - f(X_{t,y}(s))\rangle\right]\mathrm{d}s\\
&+ \int_t^{t+h}\mathbb{E}\left[\|g(X_{t,x}(s)) - g(X_{t,y}(s))\|_F^2\right]\mathrm{d}s.
\end{aligned} \tag{69}$$

Bearing this in mind and noting

$$Z_{t,x,y}(s) = X_{t,x}(t+h) - X_{t,y}(s) - (x-y), \tag{70}$$

one can further use the Cauchy-Schwarz inequality and (40) to derive

$$
\begin{aligned}
&\mathbb{E}\big[|Z_{t,x,y}(t+h)|^2\big] \\
&= \int_t^{t+h} \mathbb{E}\Big[2\langle X_{t,x}(s) - X_{t,y}(s), f\left(X_{t,x}(s)\right) - f\left(X_{t,y}(s)\right)\rangle + \big\|g\left(X_{t,x}(s)\right) - g\left(X_{t,y}(s)\right)\big\|_F^2\Big]\,\mathrm{d}s \\
&\quad - 2\int_t^{t+h} \mathbb{E}\Big[\langle x - y, f\left(X_{t,x}(s)\right) - f\left(X_{t,y}(s)\right)\rangle\Big]\,\mathrm{d}s \\
&\leq \underbrace{2L^* \int_t^{t+h} \mathbb{E}\big[|X_{t,x}(s) - X_{t,y}(s)|^2\big]\,\mathrm{d}s}_{=:I_1} + \underbrace{2|x - y| \int_t^{t+h} \mathbb{E}\big[|f\left(X_{t,x}(s)\right) - f\left(X_{t,y}(s)\right)|\big]\,\mathrm{d}s}_{=:I_2}\,.
\end{aligned}
\tag{71}
$$

In what follows, we handle the two terms in (71) separately. We first emply (67) as well as $h \leq \frac{1}{2L^*}$ to show

$$
\mathbb{E}\big[|X_{t,x}(s) - X_{t,y}(s)|^2\big] \leq e^{2L^* h}|x - y|^2 \leq e|x - y|^2 \leq 3|x - y|^2,
\tag{72}
$$

which directly implies

$$
I_1 \leq 6L^*|x - y|^2 h.
\tag{73}
$$

Before coming to the estimate of $I_2$, one can derive from (41) and the Hölder inequality that

$$
\begin{aligned}
\mathbb{E}\big[|f\left(X_{t,x}(s)\right) - f\left(X_{t,y}(s)\right)|\big] &\leq L_f^* \mathbb{E}\Big[\big(1 + |X_{t,x}(s)|^{r_0} + |X_{t,y}(s)|^{r_0}\big)|X_{t,x}(s) - X_{t,y}(s)|\Big] \\
&\leq L_f^* \Big(\mathbb{E}\big[\big(1 + |X_{t,x}(s)|^{r_0} + |X_{t,y}(s)|^{r_0}\big)^2\big]\Big)^{1/2} \Big(\mathbb{E}\big[|X_{t,x}(s) - X_{t,y}(s)|^2\big]\Big)^{1/2} \\
&\leq 3L_f^* \Big(1 + \mathbb{E}\big[|X_{t,x}(s)|^{2r_0}\big] + \mathbb{E}\big[|X_{t,y}(s)|^{2r_0}\big]\Big)^{1/2}|x - y|,
\end{aligned}
\tag{74}
$$

where we used (72) in the last step. By noting

$$
\mathbb{E}\big[|X_{t,x}(s)|^{2r_0}\big] = \mathbb{E}\big[|X_{t,y}(s)|^{2r_0}\big] = 1, \quad r_0 = 0,
\tag{75}
$$

and

$$
\mathbb{E}\big[|X_{t,x}(s)|^{2r_0}\big] \leq \big(\mathbb{E}\big[|X_{t,x}(s)|^2\big]\big)^{r_0}, \quad \mathbb{E}\big[|X_{t,y}(s)|^{2r_0}\big] \leq \big(\mathbb{E}\big[|X_{t,y}(s)|^2\big]\big)^{r_0}, \quad r_0 \in (0, 1),
\tag{76}
$$

one derives from Lemma 3.1 and $(a + b)^q \leq a^q + b^q, a, b \geq 0, q \in (0, 1)$ that

$$
\mathbb{E}\big[|X_{t,x}(s)|^{2r_0}\big] \leq \begin{cases} 1, & r_0 = 0, \\ C_0^*|x|^{2r_0} + (\hat{C}_0^*(1))^{r_0}, & r_0 \in (0, 1), \\ C_0^*|x|^{2r_0} + \hat{C}_0^*(r_0), & r_0 \in [1, \infty), \end{cases} \quad \mathbb{E}\big[|X_{t,y}(s)|^{2r_0}\big] \leq \begin{cases} 1, & r_0 = 0, \\ C_0^*|y|^{2r_0} + (\hat{C}_0^*(1))^{r_0}, & r_0 \in (0, 1), \\ C_0^*|y|^{2r_0} + \hat{C}_0^*(r_0), & r_0 \in [1, \infty). \end{cases}
\tag{77}
$$

Inserting this into (74), one can easily see

$$
\mathbb{E}\big[|f\left(X_{t,x}(s)\right) - f\left(X_{t,y}(s)\right)|\big] \leq 3L_f^*\big(\hat{C}_{r_0}^* + C_0^*|x|^{2r_0} + C_0^*|y|^{2r_0}\big)^{1/2}|x - y|,
\tag{78}
$$

where

$$
\hat{C}_{r_0}^* := \begin{cases} 1, & r_0 = 0, \\ 3 \vee 3(\hat{C}_0^*(1))^{r_0}, & r_0 \in (0, 1), \\ 3 \vee 3(\hat{C}_0^*(r_0)), & r_0 \in [1, \infty). \end{cases}
\tag{79}
$$

This suffices to ensure

$$
I_2 \leq 6L_f^*\big(\hat{C}_{r_0}^* + C_0^*|x|^{2r_0} + C_0^*|y|^{2r_0}\big)^{1/2}|x - y|^2 h.
\tag{80}
$$

Plugging (73) and (80) into (71) shows

$$
\mathbb{E}\big[|Z_{t,x,y}(t+h)|^2\big] \leq \Big(6L^* + 6L_f^*\big(\hat{C}_{r_0}^* + C_0^*|x|^{2r_0} + C_0^*|y|^{2r_0}\big)^{1/2}\Big)|x - y|^2 h,
\tag{81}
$$

as required. $\qquad\square$

## C. Proof of Theorem 3.3

*Proof of Theorem 3.3.* By (33), we have

$$
\begin{aligned}
X_{0,x_0'}\left(t_{k+1}\right) - Y_{0,x_0'}\left(t_{k+1}\right) &= X_{t_k,X(t_k)}\left(t_{k+1}\right) - Y_{t_k,Y_k}\left(t_{k+1}\right) \\
&= \left(X_{t_k,X(t_k)}\left(t_{k+1}\right) - X_{t_k,Y_k}\left(t_{k+1}\right)\right) + \left(X_{t_k,Y_k}\left(t_{k+1}\right) - Y_{t_k,Y_k}\left(t_{k+1}\right)\right).
\end{aligned}
\tag{82}
$$

The first difference on the right-hand side of (82) is caused by different initial data. The second difference is the one-step approximation error. Taking square and expectation on both sides gives

$$
\begin{aligned}
&\mathbb{E}\left[\left|X_{0,x_0'}\left(t_{k+1}\right) - Y_{0,x_0'}\left(t_{k+1}\right)\right|^2\right] \\
&= \underbrace{\mathbb{E}\left[\mathbb{E}\left[\left|X_{t_k,X(t_k)}\left(t_{k+1}\right) - X_{t_k,Y_k}\left(t_{k+1}\right)\right|^2\Big|\mathcal{F}_{t_k}\right]\right]}_{=:J_1} + \underbrace{\mathbb{E}\left[\mathbb{E}\left[\left|X_{t_k,Y_k}\left(t_{k+1}\right) - Y_{t_k,Y_k}\left(t_{k+1}\right)\right|^2\Big|\mathcal{F}_{t_k}\right]\right]}_{=:J_2} \\
&\quad + \underbrace{2\mathbb{E}\left[\mathbb{E}\left[\left(X_{t_k,X(t_k)}\left(t_{k+1}\right) - X_{t_k,Y_k}\left(t_{k+1}\right)\right)^T\left(X_{t_k,Y_k}\left(t_{k+1}\right) - Y_{t_k,Y_k}\left(t_{k+1}\right)\right)\Big|\mathcal{F}_{t_k}\right]\right]}_{=:J_3}.
\end{aligned}
\tag{83}
$$

In the following we cope with the above three items separately. By the conditional version of (44), we first get

$$
J_1 \le \left(1 + 4L^*h\right)\mathbb{E}\left[\left|X\left(t_k\right) - Y_k\right|^2\right].
\tag{84}
$$

In light of the conditional version of the second inequality in (46) and Condition (A2), one can easily treat $J_2$ as follows

$$
\begin{aligned}
J_2 &\le \left(\hat{K}_2^* + K_2^*\mathbb{E}\left[|Y_k|^{2r}\right]\right)h^{2p_2} \\
&\le \left(\hat{K}_2^* + K_2^*\left(\hat{C}_1^*(r) + C_1^*\mathbb{E}\left[|x_0'|^{2r}\right]\right)\right)h^{2p_2} \\
&= \left(\hat{K}_2^* + K_2^*\hat{C}_1^*(r) + K_2^*C_1^*\mathbb{E}\left[|x_0'|^{2r}\right]\right)h^{2p_2}.
\end{aligned}
\tag{85}
$$

In order to estimate $J_3$ in (83), we first note that

$$
X_{t_k,X(t_k)}\left(t_{k+1}\right) - X_{t_k,Y_k}\left(t_{k+1}\right) = X\left(t_k\right) - Y_k + Z_{t_k,X(t_k),Y_k}(t_{k+1}).
\tag{86}
$$

As a consequence,

$$
\begin{aligned}
J_3 &\le \underbrace{2\left|\mathbb{E}\left[\mathbb{E}\left[\left(X\left(t_k\right) - Y_k\right)^T\left(X_{t_k,Y_k}\left(t_{k+1}\right) - Y_{t_k,Y_k}\left(t_{k+1}\right)\right)\Big|\mathcal{F}_{t_k}\right]\right]\right|}_{=:J_{31}} \\
&\quad + \underbrace{2\left|\mathbb{E}\left[\mathbb{E}\left[Z_{t_k,X(t_k),Y_k}(t_{k+1})^T\left(X_{t_k,Y_k}\left(t_{k+1}\right) - Y_{t_k,Y_k}\left(t_{k+1}\right)\right)\Big|\mathcal{F}_{t_k}\right]\right]\right|}_{=:J_{32}}.
\end{aligned}
\tag{87}
$$

Since $X(t_k) - Y_k$ is $\mathcal{F}_{t_k}$-measurable, the conditional version of the first inequality in (46), together with the Hölder inequality and Condition (A2), leads to

$$
\begin{aligned}
J_{31} &= 2\left|\mathbb{E}\left[\left(X\left(t_k\right) - Y_k\right)^T\mathbb{E}\left[\left(X_{t_k,Y_k}\left(t_{k+1}\right) - Y_{t_k,Y_k}\left(t_{k+1}\right)\right)\Big|\mathcal{F}_{t_k}\right]\right]\right| \\
&\le 2\left(\mathbb{E}\left[\left|X\left(t_k\right) - Y_k\right|^2\right]\right)^{1/2}\left(\mathbb{E}\left[\left|\mathbb{E}\left[\left(X_{t_k,Y_k}\left(t_{k+1}\right) - Y_{t_k,Y_k}\left(t_{k+1}\right)\right)\Big|\mathcal{F}_{t_k}\right]\right|^2\right]\right)^{1/2} \\
&\le 2\left(\mathbb{E}\left[\left|X\left(t_k\right) - Y_k\right|^2\right]\right)^{1/2}\left(\hat{K}_1^* + K_1^*\mathbb{E}\left[|Y_k|^{2r}\right]\right)^{1/2}h^{p_1} \\
&\le 2\left(\mathbb{E}\left[\left|X\left(t_k\right) - Y_k\right|^2\right]\right)^{1/2}\left(\hat{K}_1^* + K_1^*\left(\hat{C}_1^*(r) + C_1^*\mathbb{E}\left[|x_0'|^{2r}\right]\right)\right)^{1/2}h^{p_1} \\
&\le \mathbb{E}\left[\left|X\left(t_k\right) - Y_k\right|^2\right]h + \left(\hat{K}_1^* + K_1^*\hat{C}_1^*(r) + K_1^*C_1^*\mathbb{E}\left[|x_0'|^{2r}\right]\right)h^{2p_1-1}.
\end{aligned}
\tag{88}
$$

To handle $J_{32}$, we employ the Hölder inequality to get

$$J_{32} \leq 2\mathbb{E}\left[\left(\mathbb{E}\left[\left|Z_{t_k,X(t_k),Y_k}(t_{k+1})\right|^2\Big|\mathcal{F}_{t_k}\right]\right)^{1/2}\left(\mathbb{E}\left[\left|X_{t_k,Y_k}(t_{k+1}) - Y_{t_k,Y_k}(t_{k+1})\right|^2\Big|\mathcal{F}_{t_k}\right]\right)^{1/2}\right]. \tag{89}$$

Here we utilize the conditional version of (45) to show that

$$\left(\mathbb{E}\left[\left|Z_{t_k,X(t_k),Y_k}(t_{k+1})\right|^2\Big|\mathcal{F}_{t_k}\right]\right)^{1/2} \leq \left(6L^* + 6L_f^*\left(\hat{C}_{r_0}^* + C_0^*|X(t_k)|^{2r_0} + C_0^*|Y_k|^{2r_0}\right)^{1/2}\right)^{1/2}|X(t_k) - Y_k|h^{1/2}. \tag{90}$$

With this at hand, using the inequality $\sqrt{a+b+c} \leq \sqrt{a} + \sqrt{b} + \sqrt{c}, a,b,c \geq 0$ and the conditional version of the second inequality in (46), one can derive from (89) that

$$\begin{aligned}
J_{32} \leq{}& 2\mathbb{E}\left[\left(6L^* + 6L_f^*\left(\hat{C}_{r_0}^* + C_0^*|X(t_k)|^{2r_0} + C_0^*|Y_k|^{2r_0}\right)^{1/2}\right)^{1/2}|X(t_k) - Y_k|\left(\hat{K}_2^* + K_2^*|Y_k|^{2r}\right)^{1/2}\right]h^{p_2+1/2}\\
\leq{}& 2\mathbb{E}\left[\left(6L^*\right)^{1/2}|X(t_k) - Y_k|\left(\hat{K}_2^* + K_2^*|Y_k|^{2r}\right)^{1/2}\right.\\
&\left.+ \left(6L_f^*\right)^{1/2}\left((\hat{C}_{r_0}^*)^{1/2} + C_0^*|X(t_k)|^{r_0} + C_0^*|Y_k|^{r_0}\right)^{1/2}|X(t_k) - Y_k|\left(\hat{K}_2^* + K_2^*|Y_k|^{2r}\right)^{1/2}\right]h^{p_2+1/2}\\
\leq{}& \left(6L^* + 6L_f^*\right)\mathbb{E}\left[|X(t_k) - Y_k|^2\right]h + \mathbb{E}\left[\hat{K}_2^* + K_2^*|Y_k|^{2r}\right]h^{2p_2}\\
&+ \mathbb{E}\left[\left((\hat{C}_{r_0}^*)^{1/2} + C_0^*|X(t_k)|^{r_0} + C_0^*|Y_k|^{r_0}\right)\left(\hat{K}_2^* + K_2^*|Y_k|^{2r}\right)\right]h^{2p_2}.
\end{aligned} \tag{91}$$

By Condition (A2), one can deduce

$$\mathbb{E}\left[\hat{K}_2^* + K_2^*|Y_k|^{2r}\right] \leq \left(\hat{K}_2^* + K_2^*\hat{C}_1^*(r) + K_2^*C_1^*\mathbb{E}\left[|x_0'|^{2r}\right]\right). \tag{92}$$

With regard to the third term in (91), by using the Young inequality, we obtain that, for any $r_1 \geq 1$,

$$\begin{aligned}
&\mathbb{E}\left[\left((\hat{C}_{r_0}^*)^{1/2} + C_0^*|X(t_k)|^{r_0} + C_0^*|Y_k|^{r_0}\right)\left(\hat{K}_2^* + K_2^*|Y_k|^{2r}\right)\right]\\
={}& (\hat{C}_{r_0}^*)^{1/2}\hat{K}_2^* + \mathbb{E}\left[C_0^*\hat{K}_2^*|X(t_k)|^{r_0} + C_0^*\hat{K}_2^*|Y_k|^{r_0} + K_2^*(\hat{C}_{r_0}^*)^{1/2}|Y_k|^{2r} + K_2^*C_0^*|X(t_k)|^{r_0}|Y_k|^{2r} + K_2^*C_0^*|Y_k|^{2r+r_0}\right]\\
\leq{}& (\hat{C}_{r_0}^*)^{1/2}\hat{K}_2^* + \frac{2r}{2r+r_0}C_0^*d^{-r_1}\left(d^{r_1}\hat{K}_2^*\right)^{\frac{2r+r_0}{2r}} + \frac{r_0}{2r+r_0}C_0^*d^{-r_1}\mathbb{E}\left[|X(t_k)|^{2r+r_0}\right] + \frac{2r}{2r+r_0}C_0^*d^{-r_1}\left(d^{r_1}\hat{K}_2^*\right)^{\frac{2r+r_0}{2r}}\\
&+ \frac{r_0}{2r+r_0}C_0^*d^{-r_1}\mathbb{E}\left[|Y_k|^{2r+r_0}\right] + \frac{r_0}{2r+r_0}K_2^*(\hat{C}_{r_0}^*)^{\frac{2r+r_0}{2r_0}} + \frac{2r}{2r+r_0}K_2^*\mathbb{E}\left[|Y_k|^{2r+r_0}\right] + \frac{r_0}{2r+r_0}K_2^*C_0^*\mathbb{E}\left[|X(t_k)|^{2r+r_0}\right]\\
&+ \frac{2r}{2r+r_0}K_2^*C_0^*\mathbb{E}\left[|Y_k|^{2r+r_0}\right] + K_2^*C_0^*\mathbb{E}\left[|Y_k|^{2r+r_0}\right]\\
={}& (\hat{C}_{r_0}^*)^{1/2}\hat{K}_2^* + \frac{4r}{2r+r_0}C_0^*d^{-r_1}\left(d^{r_1}\hat{K}_2^*\right)^{\frac{2r+r_0}{2r}} + \frac{r_0}{2r+r_0}K_2^*(\hat{C}_{r_0}^*)^{\frac{2r+r_0}{2r_0}} + \frac{r_0}{2r+r_0}C_0^*\left(d^{-r_1} + K_2^*\right)\mathbb{E}\left[|X(t_k)|^{2r+r_0}\right]\\
&+ \left(\frac{r_0}{2r+r_0}C_0^*d^{-r_1} + \frac{2r}{2r+r_0}K_2^* + \frac{2r}{2r+r_0}K_2^*C_0^* + K_2^*C_0^*\right)\mathbb{E}\left[|Y_k|^{2r+r_0}\right].
\end{aligned} \tag{93}$$

By virtue of Lemma 3.1, one can obtain that

$$\mathbb{E}\left[|X(t_k)|^{2r+r_0}\right] \leq C_0^*\mathbb{E}\left[|x_0'|^{2r+r_0}\right] + \hat{C}_0^*(2r + r_0). \tag{94}$$

In the same way, Condition (A2) implies

$$\mathbb{E}\left[|Y_k|^{2r+r_0}\right] \leq C_1^*\mathbb{E}\left[|x_0'|^{2r+r_0}\right] + \hat{C}_1^*(2r + r_0). \tag{95}$$

Plugging these two estimates into (93) gives

$$
\begin{aligned}
&\mathbb{E}\Big[\Big((\hat{C}_{r_0}^*)^{1/2} + C_0^*|X(t_k)|^{r_0} + C_0^*|Y_k|^{r_0}\Big)\Big(\hat{K}_2^* + K_2^*|Y_k|^{2r}\Big)\Big] \\
&\leq (\hat{C}_{r_0}^*)^{1/2}\hat{K}_2^* + \tfrac{4r}{2r+r_0}C_0^*d^{-r_1}\big(d^{r_1}\hat{K}_2^*\big)^{\frac{2r+r_0}{2r}} + \tfrac{r_0}{2r+r_0}K_2^*(\hat{C}_{r_0}^*)^{\frac{2r+r_0}{2r_0}} + \tfrac{r_0}{2r+r_0}C_0^*\big(d^{-r_1} + K_2^*\big)\Big(C_0^*\mathbb{E}\big[|x_0'|^{2r+r_0}\big] \\
&\quad + \hat{C}_0^*(2r+r_0)\Big) + \big(\tfrac{r_0}{2r+r_0}C_0^*d^{-r_1} + \tfrac{2r}{2r+r_0}K_2^* + \tfrac{2r}{2r+r_0}K_2^*C_0^* + K_2^*C_0^*\big)\Big(C_1^*\mathbb{E}\big[|x_0'|^{2r+r_0}\big] + \hat{C}_1^*(2r+r_0)\Big) \\
&= (\hat{C}_{r_0}^*)^{1/2}\hat{K}_2^* + \tfrac{4r}{2r+r_0}C_0^*d^{-r_1}\big(d^{r_1}\hat{K}_2^*\big)^{\frac{2r+r_0}{2r}} + \tfrac{r_0}{2r+r_0}K_2^*(\hat{C}_{r_0}^*)^{\frac{2r+r_0}{2r_0}} + \tfrac{r_0}{2r+r_0}C_0^*\big(d^{-r_1} + K_2^*\big)\hat{C}_0^*(2r+r_0) \\
&\quad + \big(\tfrac{r_0}{2r+r_0}C_0^*d^{-r_1} + \tfrac{2r}{2r+r_0}K_2^* + \tfrac{2r}{2r+r_0}K_2^*C_0^* + K_2^*C_0^*\big)\hat{C}_1^*(2r+r_0) + \big(\tfrac{r_0}{2r+r_0}(C_0^*)^2d^{-r_1} + \tfrac{r_0}{2r+r_0}K_2^*(C_0^*)^2 \\
&\quad + \tfrac{r_0}{2r+r_0}C_0^*C_1^*d^{-r_1} + \tfrac{2r}{2r+r_0}K_2^*C_1^* + \tfrac{2r}{2r+r_0}K_2^*C_0^*C_1^* + K_2^*C_0^*C_1^*\big)\mathbb{E}\big[|x_0'|^{2r+r_0}\big].
\end{aligned}
\tag{96}
$$

Inserting this and (92) into (91) directly implies

$$
\begin{aligned}
J_{32} \leq &\big(6L^* + 6L_f^*\big)\mathbb{E}\Big[\big|X(t_k) - Y_k\big|^2\Big]h + \Big(\hat{K}_2^* + K_2^*\hat{C}_1^*(r) + K_2^*C_1^*\mathbb{E}\big[|x_0'|^{2r}\big]\Big)h^{2p_2} + \Big((\hat{C}_{r_0}^*)^{1/2}\hat{K}_2^* \\
&+ \tfrac{4r}{2r+r_0}C_0^*d^{-r_1}\big(d^{r_1}\hat{K}_2^*\big)^{\frac{2r+r_0}{2r}} + \tfrac{r_0}{2r+r_0}K_2^*(\hat{C}_{r_0}^*)^{\frac{2r+r_0}{2r_0}} + \tfrac{r_0}{2r+r_0}C_0^*\big(d^{-r_1} + K_2^*\big)\hat{C}_0^*(2r+r_0) \\
&+ \big(\tfrac{r_0}{2r+r_0}C_0^*d^{-r_1} + \tfrac{2r}{2r+r_0}K_2^* + \tfrac{2r}{2r+r_0}K_2^*C_0^* + K_2^*C_0^*\big)\hat{C}_1^*(2r+r_0) + \big(\tfrac{r_0}{2r+r_0}(C_0^*)^2d^{-r_1} \\
&+ \tfrac{r_0}{2r+r_0}K_2^*(C_0^*)^2 + \tfrac{r_0}{2r+r_0}C_0^*C_1^*d^{-r_1} + \tfrac{2r}{2r+r_0}K_2^*C_1^* + \tfrac{2r}{2r+r_0}K_2^*C_0^*C_1^* + K_2^*C_0^*C_1^*\big)\mathbb{E}\big[|x_0'|^{2r+r_0}\big]\Big)h^{2p_2}.
\end{aligned}
\tag{97}
$$

Putting estimates of $J_{31}$ and $J_{32}$ together, we derive from (87) that

$$
\begin{aligned}
J_3 \leq &\big(1 + 6L^* + 6L_f^*\big)\mathbb{E}\big[|X(t_k) - Y_k|^2\big]h + \Big(\hat{K}_1^* + \hat{K}_2^* + K_1^*\hat{C}_1^*(r) + K_2^*\hat{C}_1^*(r) + \big(K_1^* + K_2^*\big)C_1^*\mathbb{E}\big[|x_0'|^{2r}\big]\Big)h^{2p_2} \\
&+ \Big((\hat{C}_{r_0}^*)^{1/2}\hat{K}_2^* + \tfrac{4r}{2r+r_0}C_0^*d^{-r_1}\big(d^{r_1}\hat{K}_2^*\big)^{\frac{2r+r_0}{2r}} + \tfrac{r_0}{2r+r_0}K_2^*(\hat{C}_{r_0}^*)^{\frac{2r+r_0}{2r_0}} + \tfrac{r_0}{2r+r_0}C_0^*\big(d^{-r_1} + K_2^*\big)\hat{C}_0^*(2r+r_0) \\
&+ \big(\tfrac{r_0}{2r+r_0}C_0^*d^{-r_1} + \tfrac{2r}{2r+r_0}K_2^* + \tfrac{2r}{2r+r_0}K_2^*C_0^* + K_2^*C_0^*\big)\hat{C}_1^*(2r+r_0) + \big(\tfrac{r_0}{2r+r_0}(C_0^*)^2d^{-r_1} \\
&+ \tfrac{r_0}{2r+r_0}K_2^*(C_0^*)^2 + \tfrac{r_0}{2r+r_0}C_0^*C_1^*d^{-r_1} + \tfrac{2r}{2r+r_0}K_2^*C_1^* + \tfrac{2r}{2r+r_0}K_2^*C_0^*C_1^* + K_2^*C_0^*C_1^*\big)\mathbb{E}\big[|x_0'|^{2r+r_0}\big]\Big)h^{2p_2}.
\end{aligned}
\tag{98}
$$

Collecting estimates of $J_1$, $J_2$ and $J_3$ together and noting $p_1 \geq p_2 + 1/2$, we immediately derive from (83) that

$$
\begin{aligned}
\mathbb{E}\big[|X(t_{k+1}) - Y_{k+1}|^2\big] \leq &\Big(1 + \big(1 + 10L^* + 6L_f^*\big)h\Big)\mathbb{E}\big[|X(t_k) - Y_k|^2\big] + \Big(\hat{K}_1^* + 2\hat{K}_2^* + K_1^*\hat{C}_1^*(r) + 2K_2^*\hat{C}_1^*(r) \\
&+ \big(K_1^* + 2K_2^*\big)C_1^*\mathbb{E}\big[|x_0'|^{2r}\big]\Big)h^{2p_2} + \Big((\hat{C}_{r_0}^*)^{1/2}\hat{K}_2^* + \tfrac{4r}{2r+r_0}C_0^*d^{-r_1}\big(d^{r_1}\hat{K}_2^*\big)^{\frac{2r+r_0}{2r}} \\
&+ \tfrac{r_0}{2r+r_0}K_2^*(\hat{C}_{r_0}^*)^{\frac{2r+r_0}{2r_0}} + \tfrac{r_0}{2r+r_0}C_0^*\big(d^{-r_1} + K_2^*\big)\hat{C}_0^*(2r+r_0) + \big(\tfrac{r_0}{2r+r_0}C_0^*d^{-r_1} + \tfrac{2r}{2r+r_0}K_2^* \\
&+ \tfrac{2r}{2r+r_0}K_2^*C_0^* + K_2^*C_0^*\big)\hat{C}_1^*(2r+r_0) + \big(\tfrac{r_0}{2r+r_0}(C_0^*)^2d^{-r_1} + \tfrac{r_0}{2r+r_0}K_2^*(C_0^*)^2 \\
&+ \tfrac{r_0}{2r+r_0}C_0^*C_1^*d^{-r_1} + \tfrac{2r}{2r+r_0}K_2^*C_1^* + \tfrac{2r}{2r+r_0}K_2^*C_0^*C_1^* + K_2^*C_0^*C_1^*\big)\mathbb{E}\big[|x_0'|^{2r+r_0}\big]\Big)h^{2p_2}.
\end{aligned}
\tag{99}
$$

Again, using the Hölder inequality and the Young inequality yields

$$
\mathbb{E}\big[|x_0'|^{2r}\big] \leq \big(\mathbb{E}\big[|x_0'|^{2r+r_0}\big]\big)^{\frac{2r}{2r+r_0}} \leq \tfrac{r_0}{2r+r_0} + \tfrac{2r}{2r+r_0}\mathbb{E}\big[|x_0'|^{2r+r_0}\big], \quad r_0 \geq 0.
\tag{100}
$$

Keeping this in mind and by setting

$$
e_k^2 := \mathbb{E}[|X(t_k) - Y_k|^2], \quad \kappa^* := 1 + 10L^* + 6L_f^*,
\tag{101}
$$

we thus get

$$
e_{k+1}^2 \leq \big(1 + \kappa^*h\big)e_k^2 + \big(\hat{K}^* + K^*\mathbb{E}\big[|x_0|^{2r+r_0}\big]\big)h^{2p_2},
\tag{102}
$$

where for any $r_1 \geq 1$,

$$
\begin{aligned}
\hat{K}^* :=& \hat{K}_1^* + 2\hat{K}_2^* + K_1^* \hat{C}_1^*(r) + 2K_2^* \hat{C}_1^*(r) + \tfrac{r_0}{2r+r_0}\big(K_1^* + 2K_2^*\big)C_1^* + (\hat{C}_{r_0}^*)^{1/2}\hat{K}_2^* \\
& + \tfrac{4r}{2r+r_0}C_0^* d^{-r_1}\big(d^{r_1}\hat{K}_2^*\big)^{\frac{2r+r_0}{2r}} + \tfrac{r_0}{2r+r_0}K_2^*(\hat{C}_{r_0}^*)^{\frac{2r+r_0}{2r_0}} + \tfrac{r_0}{2r+r_0}C_0^*\big(d^{-r_1} + K_2^*\big)\hat{C}_0^*(2r+r_0) \\
& + \big(\tfrac{r_0}{2r+r_0}C_0^* d^{-r_1} + \tfrac{2r}{2r+r_0}K_2^* + \tfrac{2r}{2r+r_0}K_2^* C_0^* + K_2^* C_0^*\big)\hat{C}_1^*(2r+r_0),
\end{aligned}
$$

$$
\begin{aligned}
K^* :=& \tfrac{2r}{2r+r_0}\big(K_1^* + 2K_2^*\big)C_1^* + \tfrac{r_0}{2r+r_0}(C_0^*)^2 d^{-r_1} + \tfrac{r_0}{2r+r_0}K_2^*(C_0^*)^2 + \tfrac{r_0}{2r+r_0}C_0^* C_1^* d^{-r_1} \\
& + \tfrac{2r}{2r+r_0}K_2^* C_1^* + \tfrac{2r}{2r+r_0}K_2^* C_0^* C_1^* + K_2^* C_0^* C_1^*.
\end{aligned}
\tag{103}
$$

Noting $e_0 = 0$ and by iteration one can obtain the desired assertion. $\qquad\square$

## D. Proof of Theorem 3.4

Before proving Theorem 3.4, we quote a lemma from (Ye & Zhou, 2024).

**Lemma D.1.** *Given $n_1 \in \mathbb{N}$, $\varepsilon_3 > 0$, and $q \in (0,1)$. If a non-negative sequence $\{a_n\}_{n \in \mathbb{N}_0}$ satisfies*

$$
a_n \leq \varepsilon_3 + q a_{n-n_1},
\tag{104}
$$

*for $n_1 \leq n \in \mathbb{N}$, then*

$$
a_n \leq \tfrac{\varepsilon_3}{1-q} + c_0 q^{\frac{n}{n_1}-1}, \quad \forall n \in \mathbb{N}_0,
\tag{105}
$$

*where $c_0 = \max_{i \in [n_1-1]_0}\{a_i\}$.*

The proof of this lemma can be found in Lemma 3.18 of (Ye & Zhou, 2024).

*Proof of Theorem 3.4.* By using the triangle inequality, we obtain that, for any given $n \geq n_1$,

$$
\mathcal{W}_2\big(\nu' \tilde{P}_n, \Pi\big) \leq \mathcal{W}_2\big(\nu' \tilde{P}_{n-n_1}\tilde{P}_{n_1}, \nu' \tilde{P}_{n-n_1}P_{t_{n_1}}\big) + \mathcal{W}_2\big(\nu' \tilde{P}_{n-n_1}P_{t_{n_1}}, \Pi\big).
\tag{106}
$$

With regard to $\mathcal{W}_2(\nu' \tilde{P}_{n-n_1}P_{t_{n_1}}, \Pi)$, by using Condition (A5), we derive

$$
\mathcal{W}_2\big(\nu' \tilde{P}_{n-n_1}P_{t_{n_1}}, \Pi\big) \leq \mathcal{K}^* e^{-\eta^* n_1 h}\mathcal{W}_2\big(\nu' \tilde{P}_{n-n_1}, \Pi\big), \quad \forall n \geq n_1.
\tag{107}
$$

For a given stepsize $h > 0$, we choose

$$
n_1 := \big\lceil \tfrac{\log \mathcal{K}^* + 1}{\eta^* h}\big\rceil \geq \tfrac{\log \mathcal{K}^* + 1}{\eta^* h},
\tag{108}
$$

such that

$$
\mathcal{K}^* e^{-\eta^* n_1 h} \leq \tfrac{1}{e}.
\tag{109}
$$

As a result,

$$
\mathcal{W}_2\big(\nu' \tilde{P}_{n-n_1}P_{t_{n_1}}, \Pi\big) \leq \tfrac{1}{e}\mathcal{W}_2\big(\nu' \tilde{P}_{n-n_1}, \Pi\big), \quad \forall n \geq n_1.
\tag{110}
$$

Next we treat the first term in (106). Noting that

$$
\nu' \tilde{P}_{n-n_1}\tilde{P}_{n_1} = \mathcal{L}\big(Y(0, Y_{n-n_1}; t_{n_1})\big), \quad \nu' \tilde{P}_{n-n_1}P_{t_{n_1}} = \mathcal{L}\big(X(0, Y_{n-n_1}; t_{n_1})\big),
\tag{111}
$$

recalling the definition of $\mathcal{W}_2$-distance and employing Theorem 3.3 as well as Condition (A2) lead us to, for any $n \geq n_1$,

$$
\begin{aligned}
\mathcal{W}_2^2\big(\nu' \tilde{P}_{n-n_1}\tilde{P}_{n_1}, \nu' \tilde{P}_{n-n_1}P_{t_{n_1}}\big) \leq & \mathbb{E}\big[\big|X(0, Y_{n-n_1}; t_{n_1}) - Y(0, Y_{n-n_1}; t_{n_1})\big|^2\big] \\
\leq & (C^*(t_{n_1}))^2\Big(\hat{K}^* + K^*\mathbb{E}\big[|Y_{n-n_1}|^{2r+r_0}\big]\Big)h^{2p_2-1} \\
\leq & (C^*(t_{n_1}))^2\Big(\hat{K}^* + K^*\big(\hat{C}_1^*(r + \tfrac{r_0}{2}) + C_1^*\mathbb{E}\big[|x_0'|^{2r+r_0}\big]\big)\Big)h^{2p_2-1} \\
= & (C^*(t_{n_1}))^2\Big(\hat{K}^* + K^*\hat{C}_1^*(r + \tfrac{r_0}{2}) + K^* C_1^*\mathbb{E}\big[|x_0'|^{2r+r_0}\big]\Big)h^{2p_2-1},
\end{aligned}
\tag{112}
$$

which directly implies, for any $n \geq n_1$,

$$\mathcal{W}_2\big(\nu'\tilde{P}_{n-n_1}\tilde{P}_{n_1}, \nu'\tilde{P}_{n-n_1}P_{t_{n_1}}\big) \leq C^*(t_{n_1})\Big(\hat{K}^* + K^*\hat{C}_1^*(r + \tfrac{r_0}{2}) + K^*C_1^*\mathbb{E}\big[|x_0'|^{2r+r_0}\big]\Big)^{1/2} h^{p_2-1/2}. \tag{113}$$

As $h \leq \frac{1}{2L^*}$, one can derive that

$$t_{n_1} = n_1 h \leq \big(\tfrac{\log \mathcal{K}^*+1}{\eta^* h} + 1\big)h \leq \tfrac{\log \mathcal{K}^*+1}{\eta^*} + \tfrac{1}{2L^*} =: \chi^*. \tag{114}$$

Recalling (48), one can easily see that $C^*(t_{n_1}) \leq C^*(\chi^*)$, and therefore, for any $n \geq n_1$,

$$\mathcal{W}_2\big(\nu'\tilde{P}_{n-n_1}\tilde{P}_{n_1}, \nu'\tilde{P}_{n-n_1}P_{t_{n_1}}\big) \leq C^*(\chi^*)\Big(\hat{K}^* + K^*\hat{C}_1^*(r + \tfrac{r_0}{2}) + K^*C_1^*\mathbb{E}\big[|x_0'|^{2r+r_0}\big]\Big)^{1/2} h^{p_2-1/2}. \tag{115}$$

Putting (110) and (115) together, we derive from (106) that, for any $n \geq n_1$,

$$\mathcal{W}_2\big(\nu'\tilde{P}_n, \Pi\big) \leq C^*(\chi^*)\Big(\hat{K}^* + K^*\hat{C}_1^*(r + \tfrac{r_0}{2}) + K^*C_1^*\mathbb{E}\big[|x_0'|^{2r+r_0}\big]\Big)^{1/2} h^{p_2-1/2} + \tfrac{1}{e}\mathcal{W}_2\big(\nu'\tilde{P}_{n-n_1}, \Pi\big). \tag{116}$$

Utilizing Lemma D.1 acquires, for any $n \in \mathbb{N}_0$,

$$\mathcal{W}_2\big(\nu'\tilde{P}_n, \Pi\big) \leq 2C^*(\chi^*)\Big(\hat{K}^* + K^*\hat{C}_1^*(r + \tfrac{r_0}{2}) + K^*C_1^*\mathbb{E}\big[|x_0'|^{2r+r_0}\big]\Big)^{1/2} h^{p_2-1/2} + e^{1-\frac{n}{n_1}} \sup_{k \in [n_1-1]_0} \mathcal{W}_2\big(\nu'\tilde{P}_k, \Pi\big). \tag{117}$$

Recalling the definition of $\mathcal{W}_2$-distance and using Lemma 3.1 and Condition (A2) lead to

$$\sup_{k \in [n_1-1]_0} \mathcal{W}_2\big(\nu'\tilde{P}_k, \Pi\big) \leq \Big(2 \sup_{k \in \mathbb{N}_0} \mathbb{E}\big[|Y_k|^2\big] + 2\int_{\mathbb{R}^d} |x|^2 \Pi(\mathrm{d}x)\Big)^{1/2} \leq \Big(2\big(\hat{C}_0^*(1) + \hat{C}_1^*(1)\big) + 2\big(C_0^* + C_1^*\big)\mathbb{E}\big[|x_0'|^2\big]\Big)^{1/2}. \tag{118}$$

According to (108), one can easily see that

$$\tfrac{n}{n_1} \geq \tfrac{n}{\frac{\log \mathcal{K}^*+1}{\eta^* h}+1} \geq \tfrac{\eta^* nh}{\log \mathcal{K}^*+1+\frac{\eta^*}{2L^*}} =: \lambda^* nh, \tag{119}$$

implying

$$e^{-\frac{n}{n_1}} \leq e^{-\lambda^* nh}. \tag{120}$$

Plugging this and (118) into (117) yields

$$\mathcal{W}_2\big(\nu'\tilde{P}_n, \Pi\big) \leq \hat{\mathcal{K}}_1^* h^{p_2-1/2} + \hat{\mathcal{K}}_2^* e^{-\lambda^* nh}, \quad \forall n \geq 0, \tag{121}$$

where

$$\begin{aligned}
\hat{\mathcal{K}}_1^* &:= 2C^*(\chi^*)\Big(\hat{K}^* + K^*\hat{C}_1^*(r + \tfrac{r_0}{2}) + K^*C_1^*\mathbb{E}\big[|x_0'|^{2r+r_0}\big]\Big)^{1/2}, \\
\hat{\mathcal{K}}_2^* &:= e\Big(2\big(\hat{C}_0^*(1) + \hat{C}_1^*(1)\big) + 2\big(C_0^* + C_1^*\big)\mathbb{E}\big[|x_0'|^2\big]\Big)^{1/2}.
\end{aligned} \tag{122}$$

We thus get the desired assertion. $\qquad \square$

## E. Proofs of Main Results for LMC

In this section, we list auxiliary lemmas which will be used to prove our main results. We now carry out the error analysis of LMC (2). The one-step LMC approximation $\bar{Y}$ is defined by

$$\bar{Y}(t, x; t+h) := x - \nabla U(x)h + \sqrt{2}(W_{t+h} - W_t). \tag{123}$$

Equipped with Assumptions 2.6, 2.8, one can prove one-step strong and weak errors as follows.

**Lemma E.1** (One-step errors analysis of LMC). *Let Assumptions 2.1, 2.6, 2.8 hold and let $X(t, h; t+h)$ denote the solution to the Langevin SDE (1) at $t+h$, starting from the initial value $x$ at $t$. Then the one-step LMC (123) has local weak and strong errors of order 2 and 1.5, respectively, i.e., for any $t \geq 0$, $0 < h < 1$ and $x \in \mathbb{R}^d$,*

$$\left|\mathbb{E}\big[X(t,x;t+h) - \bar{Y}(t,x;t+h)\big]\right| \leq \big(\hat{K}_{\bar{1}} + K_{\bar{1}}|x|^2\big)^{1/2} h^2,$$

$$\left(\mathbb{E}\big[\big|X(t,x;t+h) - \bar{Y}(t,x;t+h)\big|^2\big]\right)^{1/2} \leq \big(\hat{K}_{\bar{2}} + K_{\bar{2}}|x|^2\big)^{1/2} h^{\frac{3}{2}}, \tag{124}$$

*where*

$$\hat{K}_{\bar{1}} := 4\big(\tfrac{2+2\mu'}{c}L_0^2 + L_0'^2 + \tfrac{2+2\mu'}{c}L_1^4 + L_1^2 L_1'^2\big)d, \qquad K_{\bar{1}} := 4(L_0^2 + L_1^4), \tag{125}$$

$$\hat{K}_{\bar{2}} := 4\big(\tfrac{2+2\mu'}{c}L_1^4 + 2L_1^2 L_1'^2 + L_1^2\big)d, \qquad K_{\bar{2}} := 4L_1^4. \tag{126}$$

*Proof.* We first notice that

$$X(t,x;t+h) - \bar{Y}(t,x;t+h) = -\int_t^{t+h} \big(\nabla U(X(t,x;s)) - \nabla U(x)\big)\,\mathrm{d}s, \tag{127}$$

where the Itô formula gives

$$\nabla U(X(t,x;s)) = \nabla U(x) - \int_t^s \big(\nabla^2 U(X(t,x;r))\nabla U(X(t,x;r)) + \nabla(\Delta(U(X(t,x;r))))\big)\,\mathrm{d}r$$

$$+ \sqrt{2}\int_t^s \nabla^2 U(X(t,x;r))\,\mathrm{d}W_r. \tag{128}$$

Then taking expectation leads to

$$\left|\mathbb{E}\big[X(t,x;t+h) - \bar{Y}(t,x;t+h)\big]\right| = \left|\mathbb{E}\Big[\int_t^{t+h}\int_t^s \big(\nabla^2 U(X(t,x;r))\nabla U(X(t,x;r)) + \nabla(\Delta(U(X(t,x;r))))\big)\,\mathrm{d}r\mathrm{d}s\Big]\right|$$

$$\leq \int_t^{t+h}\int_t^s \mathbb{E}\big[\big|\nabla^2 U(X(t,x;r))\nabla U(X(t,x;r)) + \nabla(\Delta(U(X(t,x;r))))\big|\big]\,\mathrm{d}r\mathrm{d}s. \tag{129}$$

Thanks to (10), (13), Assumption 2.8 and Lemma 2.4, we arrive at

$$\left|\mathbb{E}\big[X(t,x;t+h) - \bar{Y}(t,x;t+h)\big]\right|$$

$$\leq \int_t^{t+h}\int_t^s \Big(2\mathbb{E}\big[\big|\nabla^2 U(X(t,x;r))\nabla U(X(t,x;r))\big|^2 + \big|\nabla(\Delta(U(X(t,x;r))))\big|^2\big]\Big)^{1/2}\,\mathrm{d}r\mathrm{d}s$$

$$\leq \int_t^{t+h}\int_t^s \Big(2\mathbb{E}\big[2L_1^2 L_1'^2 d + 2L_1^4|X(t,x;r)|^2 + 2L_0'^2 d + 2L_0^2|X(t,x;r)|^2\big]\Big)^{1/2}\,\mathrm{d}r\mathrm{d}s$$

$$\leq \int_t^{t+h}\int_t^s \Big(4(L_0^2 + L_1^4)\mathbb{E}\big[|X(t,x;r)|^2\big] + 4(L_0'^2 + L_1^2 L_1'^2)d\Big)^{1/2}\,\mathrm{d}r\mathrm{d}s \tag{130}$$

$$\leq \Big(4(L_0^2 + L_1^4)(\tfrac{2+2\mu'}{c}d + |x|^2) + 4(L_1^2 L_1'^2 + L_0'^2)d\Big)^{1/2} h^2$$

$$= \Big(4\big(\tfrac{2+2\mu'}{c}L_0^2 + L_0'^2 + \tfrac{2+2\mu'}{c}L_1^4 + L_1^2 L_1'^2\big)d + 4(L_0^2 + L_1^4)|x|^2\Big)^{1/2} h^2.$$

The first assertion of (124) is thus validated. Recalling (127), one can use the Hölder inequality, the Lipschitz condition (9) to derive

$$\mathbb{E}\big[\big|X(t,x;t+h) - \bar{Y}(t,x;t+h)\big|^2\big] \leq h\int_t^{t+h} \mathbb{E}\big[\big|\nabla U(X(t,x;s)) - \nabla U(x)\big|^2\big]\,\mathrm{d}s$$

$$\leq L_1^2 h\int_t^{t+h} \mathbb{E}\big[\big|X(t,x;s) - x\big|^2\big]\,\mathrm{d}s, \tag{131}$$

where

$$X(t,x;s) - x = \int_t^s -\nabla U(X_{t,x}(r))\,\mathrm{d}r + \int_t^s \sqrt{2}\,\mathrm{d}W_r. \tag{132}$$

As a result, using (10) yields, for any $s \in [t, t+h]$,

$$
\begin{aligned}
\mathbb{E}\big[|X(t,x;s) - x|^2\big] &\leq 2\bigg(\mathbb{E}\bigg[\bigg|\int_t^s -\nabla U(X_{t,x}(r))\,\mathrm{d}r\bigg|^2\bigg] + \mathbb{E}\bigg[\bigg|\int_t^s \sqrt{2}\,\mathrm{d}W_r\bigg|^2\bigg]\bigg) \\
&\leq 2h\int_t^s \mathbb{E}\big[|\nabla U(X_{t,x}(r))|^2\big]\,\mathrm{d}r + 4dh \\
&\leq 4L_1^2 h\int_t^s \mathbb{E}\big[|X_{t,x}(r)|^2\big]\,\mathrm{d}r + 4L_1^{'2}dh + 4dh \\
&\leq \big(4(\tfrac{2+2\mu'}{c}L_1^2 + L_1^{'2} + 1)d + 4L_1^2|x|^2\big)h.
\end{aligned}
\tag{133}
$$

Keeping this in mind, we derive from (131) that

$$
\mathbb{E}\big[|X(t,x;t+h) - \bar{Y}(t,x;t+h)|^2\big] \leq \big(4(\tfrac{2+2\mu'}{c}L_1^4 + 2L_1^2 L_1^{'2} + L_1^2)d + 4L_1^4|x|^2\big)h^3,
\tag{134}
$$

as required. $\qquad\square$

Armed with Lemma E.1 and by verifying all Conditions (A1)–(A4) for the Langevin dynamics (1), one can derive the following error bound of LMC in finite time from Theorem 3.3.

**Lemma E.2** (Error analysis of LMC in finite time). *Let Assumptions 2.1, 2.6, 2.8 hold and let $\{X_t\}_{t \geq 0}$ denote the solution of the Langevin SDE* (1). *If the timestep satisfies $h \leq \frac{1}{2L} \wedge \frac{\mu}{4L_1^2} \wedge \frac{1}{\mu} \wedge 1$, then for a fixed $n_1 \in \mathbb{N}$ and denoting $T := n_1 h$, the LMC* (2) *has global mean-square error of order one in finite time, i.e.,*

$$
\sup_{n \in [n_1]} \Big(\mathbb{E}\big[|X_{t_n} - \bar{Y}_n|^2\big]\Big)^{1/2} \leq \bar{C}(T)\big(\hat{K}_L + K_L\mathbb{E}[|x_0|^2]\big)^{1/2}h,
\tag{135}
$$

*where $\bar{C}(T)$ and $\hat{K}_L$, $K_L$ are given by* (143) *and* (144), *respectively.*

*Proof.* To apply Theorem 3.3, we have to verify all conditions imposed there. By Assumption 2.1, Condition (A1) holds true with

$$
\hat{\mu}^* = \mu'd + (2p^*-1)d, \quad \mu^* = \mu
\tag{136}
$$

for any $p^* \geq 1$. Recalling Lemma 2.9, one can easily see that Condition (A2) is fulfilled with

$$
h_0 = \tfrac{\mu}{4L_1^2} \wedge \tfrac{1}{\mu} \wedge 1, \quad C_1^* := 1 \geq e^{-\mu t_n}, \quad \hat{C}_1^*(1) = \tfrac{4+4L_1^{'2}+2\mu'}{\mu}d.
\tag{137}
$$

By Assumption 2.2, Condition (A3) holds true with

$$
L^* = L.
\tag{138}
$$

Similarly, Assumption 2.6 implies that Condition (A4) is satisfied with

$$
r_0 = 0, \quad L_f^* = \tfrac{1}{3}L_1.
\tag{139}
$$

Moreover, by Lemma 2.4, we obtain that Lemma 3.1 holds true with

$$
C_0^* := 1 \geq e^{-cpt} \quad \hat{C}_0^*(p) = \tfrac{2(2p-1+\mu')^p}{cp}\big(\tfrac{2p-2}{(2\mu-c)p}\big)^{p-1}d^p.
\tag{140}
$$

In light of Lemma E.1, one can verify that (46) in Theorem 3.3 holds true with

$$
r = 1, \quad r_1 = 0, \quad \hat{K}_1^* = \hat{K}_{\bar{1}}, \quad K_1^* = K_{\bar{1}}, \quad p_1 = 2, \quad \hat{K}_2^* = \hat{K}_{\bar{2}}, \quad K_2^* = \hat{K}_{\bar{2}}, \quad p_2 = \tfrac{3}{2}.
\tag{141}
$$

All in all, one can derive from Theorem 3.3 that

$$
\Big(\mathbb{E}\big[|X_{t_n} - \bar{Y}_n|^2\big]\Big)^{1/2} \leq e^{\frac{1}{2}(1+10L+2L_1)T}\Big(\big(\hat{K}_{\bar{1}} + 5\hat{K}_{\bar{2}}\big) + \tfrac{(4+4L_1^{'2}+2\mu')d}{\mu}\big(K_{\bar{1}} + 5K_{\bar{2}}\big) + \big(K_{\bar{1}} + 5K_{\bar{2}}\big)\mathbb{E}[|x_0|^2]\Big)^{1/2}h.
\tag{142}
$$

As a consequence, the desired assertion (135) is validated by setting

$$\bar{C}(T) := e^{\frac{1}{2}(1+10L+2L_1)T},$$ (143)

and

$$
\begin{aligned}
\hat{K}_L :=& \left(\hat{K}_{\bar{1}} + 5\hat{K}_{\bar{2}}\right) + \frac{(4+4L_1'^2+2\mu')d}{\mu}\left(K_{\bar{1}} + 5K_{\bar{2}}\right) \\
=& \left(4\left(\frac{2+2\mu'}{c}L_0^2 + L_0'^2 + \frac{2+2\mu'}{c}L_1^4 + L_1^2L_1'^2\right) + 20\left(\frac{2+2\mu'}{c}L_1^4 + 2L_1^2L_1'^2 + L_1^2\right) + \frac{(4+4L_1'^2+2\mu')d}{\mu}\left(4L_0^2 + 24L_1^4\right)\right)d, \\
K_L :=& \left(K_{\bar{1}} + 5K_{\bar{2}}\right) = \left(4L_0^2 + 24L_1^4\right).
\end{aligned}
$$ (144)

$\square$

Next we aim to prove the main result of the LMC, by using Theorem 3.4.

*Proof of Theorem 2.10.* Except for Condition (**A5**), all conditions in Theorem 3.4 have been verified previously. In view of Proposition 2.5, Condition (**A5**) holds true with

$$\mathcal{K}^* = \mathcal{K}, \quad \eta^* = \eta.$$ (145)

By means of Theorem 3.4, we have

$$\mathcal{W}_2(\nu\bar{p}_n, \pi) \leq \hat{\mathcal{K}}_{\bar{1}}h + \hat{\mathcal{K}}_{\bar{2}}e^{-\lambda nh}, \quad \forall n \in \mathbb{N}_0,$$ (146)

where

$$
\begin{aligned}
\lambda &:= \frac{\eta}{\log\mathcal{K}+1+\eta/(2L)}, \quad \hat{\mathcal{K}}_{\bar{1}} := 2\bar{C}(\chi)\left(\hat{K}_L + \frac{4+4L_1'^2+2\mu'}{\mu}K_Ld + K_L\mathbb{E}[|x_0|^2]\right)^{1/2}, \\
\chi &:= \frac{\log\mathcal{K}+1}{\eta} + \frac{1}{2L}, \quad \hat{\mathcal{K}}_{\bar{2}} := 2e\left(\left(\frac{1+\mu'}{c} + \frac{2+2L_1'^2+\mu'}{\mu}\right)d + \mathbb{E}[|x_0|^2]\right)^{1/2}.
\end{aligned}
$$ (147)

In view of Assumption 2.7 and (143), (144), one can obtain the desired assertion (15), by setting

$$
\begin{aligned}
\bar{C}_1 :=& 2e^{\frac{1}{2}(1+10L+2L_1)\left(\frac{\log\mathcal{K}+1}{\eta} + \frac{1}{2L_1}\right)}\left(4\left(\frac{2+2\mu'}{c}L_0^2 + L_0'^2 + \frac{2+2\mu'}{c}L_1^4 + L_1^2L_1'^2\right) + 20\left(\frac{2+2\mu'}{c}L_1^4 + 2L_1^2L_1'^2 + L_1^2\right) \right. \\
& \left. + \frac{4+4L_1'^2+2\mu'}{\mu}\left(8L_0^2 + 48L_1^4\right) + \left(4L_0^2 + 24L_1^4\right)\sigma_1\right)^{1/2}, \\
\bar{C}_2 :=& 2e\left(\frac{1+\mu'}{c} + \frac{2+2L_1'^2+\mu'}{\mu} + \sigma_1\right)^{1/2}.
\end{aligned}
$$ (148)

We thus complete this proof. $\square$

*Proof of Proposition 2.11.* Given an error tolerance $\epsilon > 0$, by means of Theorem 2.10, one can choose $k$ to be large enough and $h$ to be small enough such that

$$\bar{C}_1\sqrt{d}e^{-\lambda kh} \leq \frac{\epsilon}{2}, \quad \bar{C}_2\sqrt{d}h \leq \frac{\epsilon}{2},$$ (149)

ensuing

$$\mathcal{W}_2(\nu\bar{p}_k, \pi) \leq \epsilon.$$ (150)

Solving the first term of inequality (149) shows

$$k \geq \frac{1}{\lambda h}\log\left(\frac{2\bar{C}_1\sqrt{d}}{\epsilon}\right),$$ (151)

and second part of inequality (149) shows

$$\frac{1}{h} \geq \frac{2\bar{C}_2\sqrt{d}}{\epsilon}.$$ (152)

Inserting this into (151) yields

$$k \geq \frac{1}{\lambda} \cdot \frac{2\bar{C}_2\sqrt{d}}{\epsilon} \cdot \log\left(\frac{2\bar{C}_1\sqrt{d}}{\epsilon}\right) = \tilde{O}\left(\frac{\sqrt{d}}{\epsilon}\right),$$ (153)

as claimed. $\square$

## F. Proofs of Main Results for Modified LMC

The organization of this section is similar to the previous section. Clearly, The one-step pLMC scheme is given by

$$\check{Y}(t,x;t+h) := \mathcal{T}^h(x) - \nabla U(\mathcal{T}^h(x))h + \sqrt{2}(W_{t+h} - W_t). \tag{154}$$

In the following we prove some useful properties of the pLMC algorithm (22).

**Lemma F.1.** *Let Assumption 2.12 hold and let $\mathcal{T}^h$ be defined as (23). Then, for all $x \in \mathbb{R}^d$, the following estimates hold true*

$$|\mathcal{T}^h(x)| \leq \min\left\{|x|, \vartheta d^{\frac{1}{2(\gamma+1)}} h^{-\frac{1}{2(\gamma+1)}}\right\}, \quad |\nabla U(\mathcal{T}^h(x))| \leq (\mathbb{L}_1' + 2\mathbb{L}_1)\vartheta^{\gamma+1} d^{1/2} h^{-1/2}. \tag{155}$$

The proof is straightforward, which can also be found in Lemma 3.3 (Pang et al., 2025).

**Lemma F.2.** *Let $\gamma > 0$ be given in Assumption 2.12 and let $\mathcal{T}^h$ be defined as (23). Then, for all $x \in \mathbb{R}^d$, we have*

$$|x - \mathcal{T}^h(x)| \leq 2\vartheta^{-4\gamma-4} d^{-2} h^2 |x|^{4\gamma+5}. \tag{156}$$

The proof of this lemma can be found in Lemma 5.2 of (Pang et al., 2025).

Thanks to Assumptions 2.12 and 2.14, we can prove one-step strong and weak errors as follows.

**Lemma F.3** (One-step errors analysis of pLMC). *Let Assumptions 2.1, 2.12, 2.14 hold and let $X(t, h; t+h)$ denote the solution to the Langevin SDE (1) at $t + h$, starting from the initial value $x$ at $t$. Then the one-step pLMC (154) has local weak and local strong errors of order 2 and 1.5, respectively, i.e., for any $t \geq 0$, $0 < h < 1$ and $x \in \mathbb{R}^d$,*

$$\left|\mathbb{E}\left[X(t,x;t+h) - \check{Y}(t,x;t+h)\right]\right| \leq \left(\hat{K}_{\check{1}} + K_{\check{1}}|x|^{10\gamma+10}\right)^{1/2} h^2,$$
$$\left(\mathbb{E}\left[\left|X(t,x;t+h) - \check{Y}(t,x;t+h)\right|^2\right]\right)^{1/2} \leq \left(\hat{K}_{\check{2}} + K_{\check{2}}|x|^{10\gamma+10}\right)^{1/2} h^{\frac{3}{2}}, \tag{157}$$

*where $\hat{K}_{\check{1}} := Cd^{5\gamma+1}$, $K_{\check{1}} := Cd^{-4}$, $\hat{K}_{\check{2}} := Cd^{5\gamma+1}$, $K_{\check{2}} := Cd^{-4}$ with the uniform constant $C := C(\mu, \mu', c, \gamma, \vartheta, \mathbb{L}_0, \mathbb{L}_0', \mathbb{L}_1, \mathbb{L}_1')$ not depending on $d$.*

*Proof.* Noticing that

$$X(t,x;t+h) - \check{Y}(t,x;t+h) = x - \mathcal{T}^h(x) - \int_t^{t+h} \left(\nabla U(X(t,x;s)) - \nabla U(\mathcal{T}^h(x))\right) \mathrm{d}s, \tag{158}$$

one can use the Itô formula to derive

$$X(t,x;t+h) - \check{Y}(t,x;t+h) = -\int_t^{t+h} \int_t^s \left(\nabla^2 U(X(t,x;r))\nabla U(X(t,x;r)) + \nabla(\Delta(U(X(t,x;r))))\right) \mathrm{d}r\mathrm{d}s$$
$$+ \sqrt{2}\int_t^{t+h} \int_t^s \nabla^2 U(X(t,x;r)) \mathrm{d}W_r\mathrm{d}s + \left(x - \mathcal{T}^h(x)\right) - \left(\nabla U(x) - \nabla U(\mathcal{T}^h(x))\right)h. \tag{159}$$

Taking expectation on both sides and using the Hölder inequality, Lemmas F.1, F.2, Assumption 2.12 yield

$$\left|\mathbb{E}\left[X(t,x;t+h) - \check{Y}(t,x;t+h)\right]\right|$$
$$\leq \left|\mathbb{E}\left[\int_t^{t+h} \int_t^s \nabla^2 U(X(t,x;r))\nabla U(X(t,x;r)) + \nabla(\Delta(U(X(t,x;r)))) \mathrm{d}r\mathrm{d}s\right]\right|$$
$$+ \left|x - \mathcal{T}^h(x)\right| + \left|\nabla U(x) - \nabla U(\mathcal{T}^h(x))\right|h$$
$$\leq \int_t^{t+h} \int_t^s \mathbb{E}\left[\left|\nabla^2 U(X(t,x;r))\nabla U(X(t,x;r))\right|\right] \mathrm{d}r\mathrm{d}s + \int_t^{t+h} \int_t^s \mathbb{E}\left[\left|\nabla(\Delta(U(X(t,x;r))))\right|\right] \mathrm{d}r\mathrm{d}s$$
$$+ 2\vartheta^{-4\gamma-4} d^{-2}|x|^{4\gamma+5}h^2 + \mathbb{L}_1\left(1 + |x|^\gamma + |\mathcal{T}^h(x)|^\gamma\right)\left|x - \mathcal{T}^h(x)\right|h$$
$$\leq \int_t^{t+h} \int_t^s \left(\mathbb{E}\left[\left|\nabla^2 U(X(t,x;r))\nabla U(X(t,x;r))\right|^2\right]\right)^{1/2} \mathrm{d}r\mathrm{d}s + \int_t^{t+h} \int_t^s \left(\mathbb{E}\left[\left|\nabla(\Delta(U(X(t,x;r))))\right|^2\right]\right)^{1/2} \mathrm{d}r\mathrm{d}s$$
$$+ 2\vartheta^{-4\gamma-4} d^{-2}|x|^{4\gamma+5}h^2 + 2\mathbb{L}_1\left(1 + |x|^\gamma + \vartheta^\gamma d^{\frac{\gamma}{2(\gamma+1)}} h^{-\frac{\gamma}{2(\gamma+1)}}\right)\vartheta^{-4\gamma-4} d^{-2}|x|^{4\gamma+5}h^3. \tag{160}$$

By using (18), (19), Lemma 2.4 and the Young inequality, we have

$$
\begin{aligned}
\mathbb{E}\Big[\big|\nabla^2 U(X(t,x;r))\nabla U(X(t,x;r))\big|^2\Big] &\leq \mathbb{L}_1^2 \mathbb{E}\Big[\big(1+|X(t,x;r)|^{2\gamma}\big)\big|\nabla U(X(t,x;r))\big|^2\Big] \\
&\leq 2\mathbb{L}_1^2 \mathbb{E}\Big[\big(1+|X(t,x;r)|^{2\gamma}\big)\big(\mathbb{L}_1'^2 d + 4\mathbb{L}_1^2 |X(t,x;r)|^{2\gamma+2}\big)\Big] \\
&\leq C\Big(d^{2\gamma+1} + \mathbb{E}\big[|X(t,x;r)|^{4\gamma+2}\big]\Big) \\
&\leq C\big(d^{2\gamma+1} + |x|^{4\gamma+2}\big),
\end{aligned}
\tag{161}
$$

where $C := C(\mu,\mu',c,\gamma,\mathbb{L}_1,\mathbb{L}_1')$. Similarly, one can utilize Assumption 2.14 to attain

$$
\mathbb{E}\Big[\big|\nabla(\Delta(U(X(t,x;r))))\big|^2\Big] \leq 2\big(\mathbb{L}_0'^2 d^{\gamma+1} + \mathbb{L}_0^2 d^{\gamma_1}\mathbb{E}\big[|X(t,x;r)|^{2\gamma_2}\big]\big) \leq C\big(d^{\gamma+1}+|x|^{2\gamma+1}\big),
\tag{162}
$$

where $C := C(\mu,\mu',c,\gamma,\mathbb{L}_0,\mathbb{L}_0)$. Thanks to the fact $d^{\frac{\gamma}{2(\gamma+1)}} \leq d^{\frac{\gamma}{2}}$ and the Young inequality

$$
d^{\gamma/2}|x|^{4\gamma+5} \leq \tfrac{\gamma}{5\gamma+5}d^{(5\gamma+5)/2} + \tfrac{4\gamma+5}{5\gamma+5}|x|^{5\gamma+5},
\tag{163}
$$

one can deduce

$$
\begin{aligned}
&2\vartheta^{-4\gamma-4}d^{-2}|x|^{4\gamma+5}h^2 + 2\mathbb{L}_1\Big(1+|x|^\gamma+\vartheta^\gamma d^{\frac{\gamma}{2(\gamma+1)}}h^{-\frac{\gamma}{2(\gamma+1)}}\Big)\vartheta^{-4\gamma-4}d^{-2}|x|^{4\gamma+5}h^3 \\
&\leq C\Big(d^{-2}|x|^{4\gamma+5} + d^{-2}|x|^{5\gamma+5} + d^{-2}d^{\frac{\gamma}{2(\gamma+1)}}|x|^{4\gamma+5}\Big)h^2 \\
&\leq C\big(d^{5\gamma+1} + d^{-4}|x|^{10\gamma+10}\big)^{1/2}h^2,
\end{aligned}
\tag{164}
$$

where $C := C(\gamma,\vartheta,\mathbb{L}_1,\mathbb{L}_1')$. Putting these estimates together, we derive from (160) that

$$
\big|\mathbb{E}\big[X(t,x;t+h) - \check{Y}(t,x;t+h)\big]\big| \leq C\big(d^{5\gamma+1} + d^{-4}|x|^{10\gamma+10}\big)^{1/2}h^2,
\tag{165}
$$

where $C := C(\mu,\mu',c,\gamma,\vartheta,\mathbb{L}_0,\mathbb{L}_0',\mathbb{L}_1,\mathbb{L}_1')$. The first assertion of (157) is thus comfirmed. Next, taking square and then expectation on both sides of (158), one can use Assumption 2.12, the Hölder inequality and Lemma F.2 to obtain

$$
\begin{aligned}
&\mathbb{E}\big[\big|X(t,x;t+h) - \check{Y}(t,x;t+h)\big|^2\big] \\
&\leq 2\big|x - \mathcal{T}^h(x)\big|^2 + 2\mathbb{E}\Big[\Big|\int_t^{t+h}\big(\nabla U(X(t,x;s)) - \nabla U(\mathcal{T}^h(x))\big)\,\mathrm{d}s\Big|^2\Big] \\
&\leq 4\vartheta^{-8\gamma-8}d^{-4}|x|^{8\gamma+10}h^4 + 4h\int_t^{t+h}\mathbb{E}\Big[\big|\nabla U(x) - \nabla U(\mathcal{T}^h(x))\big|^2\Big] + \mathbb{E}\Big[\big|\nabla U(X(t,x;s)) - \nabla U(x)\big|^2\Big]\,\mathrm{d}s \\
&\leq 4\vartheta^{-8\gamma-8}d^{-4}|x|^{8\gamma+10}h^4 + 16\mathbb{L}_1^2\big(1+|x|^{2\gamma}+\vartheta^{2\gamma}d^{\frac{\gamma}{\gamma+1}}h^{-\frac{\gamma}{\gamma+1}}\big)\vartheta^{-8\gamma-8}d^{-4}|x|^{8\gamma+10}h^4 \\
&\quad + 4\mathbb{L}_1 h\int_t^{t+h}\mathbb{E}\Big[\big|\big(1+|x|^\gamma+|X(t,x;s)|^\gamma\big)|X(t,x;s)-x|\big|^2\Big]\,\mathrm{d}s.
\end{aligned}
\tag{166}
$$

Using the Young inequality yields

$$
\begin{aligned}
&4\vartheta^{-8\gamma-8}d^{-4}|x|^{8\gamma+10}h^4 + 16\mathbb{L}_1^2\big(1+|x|^{2\gamma}+\vartheta^{2\gamma}d^{\frac{\gamma}{\gamma+1}}h^{-\frac{\gamma}{\gamma+1}}\big)\vartheta^{-8\gamma-8}d^{-4}|x|^{8\gamma+10}h^4 \\
&\leq C\big(d^{-4}|x|^{8\gamma+10} + d^{-4}|x|^{10\gamma+10} + d^{-4}d^{\frac{\gamma}{\gamma+1}}|x|^{8\gamma+10}\big)h^3 \\
&\leq C\big(d^{5\gamma+1} + d^{-4}|x|^{10\gamma+10}\big)h^3,
\end{aligned}
\tag{167}
$$

where $C := C(\gamma,\vartheta,\mathbb{L}_1,\mathbb{L}_1')$. Now, we estimate the third term in (166). It follows from Lemma 2.4, (18) that, for $q \geq 1$,

$$
\begin{aligned}
&\mathbb{E}\big[|X(t,x;s)-x|^{2q}\big] \\
&\leq 2^{2q-1}\bigg(\mathbb{E}\Big[\Big|\int_t^s -\nabla U(X_{t,x}(r))\,\mathrm{d}r\Big|^{2q}\Big] + \mathbb{E}\Big[\Big|\int_t^s \sqrt{2}\,\mathrm{d}W_r\Big|^{2q}\Big]\bigg) \\
&\leq 2^{2q-1}h^{2q-1}\int_t^s \mathbb{E}\big[|\nabla U(X_{t,x}(r))|^{2q}\big]\,\mathrm{d}r + 2^{3q-1}(2q-1)!!d^q h^q \\
&\leq C\big(d^q + |x|^{2q(\gamma+1)}\big)h^q,
\end{aligned}
\tag{168}
$$

where $C := C(\mu, \mu', c, \mathbb{L}_1, \mathbb{L}_1', q)$. Bearing this in mind and using the Hölder inequality and Lemma 2.4 give

$$
\begin{aligned}
&\int_t^{t+h} \mathbb{E}\Big[\big|(1 + |x|^\gamma + |X(t,x;s)|^\gamma)|X(t,x;s) - x|\big|^2\Big]\mathrm{d}s \\
&\leq \int_t^{t+h} \left(\mathbb{E}\Big[(1 + |x|^\gamma + |X(t,x;s)|^\gamma)^{\frac{4\gamma+2}{\gamma}}\Big]\right)^{\frac{\gamma}{2\gamma+1}} \left(\mathbb{E}\Big[|X(t,x;s) - x|^{\frac{4\gamma+2}{\gamma+1}}\Big]\right)^{\frac{\gamma+1}{2\gamma+1}}\mathrm{d}s \\
&\leq C\big(d^{2\gamma+1} + |x|^{4\gamma+2}\big)^{1/2}h^2,
\end{aligned}
\tag{169}
$$

where $C := C(\mu, \mu', c, \gamma, \mathbb{L}_1, \mathbb{L}_1')$. Collecting (167), (169) together, one can derive from (166) that

$$
\mathbb{E}\big[|X(t,x;t+h) - \check{Y}(t,x;t+h)|^2\big] \leq C\big(d^{5\gamma+1} + d^{-4}|x|^{10\gamma+10}\big)h^3,
\tag{170}
$$

where $C := C(\mu, \mu', c, \gamma, \vartheta, \mathbb{L}_0, \mathbb{L}_0', \mathbb{L}_1, \mathbb{L}_1')$. Now the second assertion in (157) is proved. $\qquad\square$

Equipped with Lemma F.3, one can follow a similar line to Lemma E.2 to prove the error analysis of pLMC in finite time.

**Lemma F.4** (Error analysis of pLMC in finite time). *Let Assumptions 2.1, 2.2, 2.12, 2.14 hold and let $\{X_t\}_{t\geq 0}$ denote the solution of the Langevin SDE (1). If the timestep satisfies $h \leq \frac{1}{2L} \wedge \frac{1}{2\mu} \wedge \frac{2\mu}{\mu+2(\mathbb{L}_1'+2\mathbb{L}_1)^2} \wedge 1$, then for a fixed $n_1 \in \mathbb{N}$ and denoting $T := n_1 h$, the pLMC (22) has global mean-square error of order one, i.e.,*

$$
\sup_{n \in [n_1]} \left(\mathbb{E}\big[|X_{t_n} - \check{Y}_n|^2\big]\right)^{1/2} \leq \check{C}(T)\big(\hat{K}_p + K_p \mathbb{E}[|x_0|^{11\gamma+10}]\big)^{1/2}h,
\tag{171}
$$

*where $\check{C}(T)$, $\hat{K}_p$ and $K_p$ are given by (178).*

*Proof.* Owing to Assumption 2.1, Condition (A1) holds true with

$$
\hat{\mu}^* = \mu'd + (2p^* - 1)d, \quad \mu^* = \mu
\tag{172}
$$

for any $p^* \geq 1$. In view of Lemma 2.15, Condition (A2) is fulfilled with

$$
h_0 = \frac{1}{2\mu} \wedge \frac{2\mu}{\mu+2(\mathbb{L}_1'+2\mathbb{L}_1)^2} \wedge 1, \quad C_1^* = 1 \geq e^{-\frac{\mu}{2}t}, \quad \hat{C}_1^*(p) = C_{\check{1}}d^p.
\tag{173}
$$

By Assumptions 2.2 and 2.12, Conditions (A3) and (A4) hold true with

$$
L^* = L, \quad r_0 = \gamma, \quad L_f^* = \mathbb{L}_1.
\tag{174}
$$

Additionally, taking Lemma 2.4 into account, Lemma 3.1 holds true with

$$
C_0^* := 1 \geq e^{-cpt} \quad \hat{C}_0^*(p) = \frac{2(2p-1+\mu')^p}{cp}\Big(\frac{2p-2}{(2\mu-c)p}\Big)^{p-1}d^p.
\tag{175}
$$

In light of Lemma F.4, one can verify that (46) in Theorem 3.3 holds with

$$
r = 5\gamma + 5, \quad r_1 = 4, \quad \hat{K}_1^* = \hat{K}_{\check{1}}, \quad K_1^* = K_{\check{1}}, \quad p_1 = 2, \quad \hat{K}_2^* = \hat{K}_{\check{2}}, \quad K_2^* = K_{\check{2}}, \quad p_2 = \frac{3}{2}.
\tag{176}
$$

With these arguments at hand, one can apply Theorem 3.3 to acquire

$$
\left(\mathbb{E}\big[|X_{t_n} - \check{Y}_n|^2\big]\right)^{1/2} \leq e^{\frac{1}{2}(1+10L+6\mathbb{L}_1)T}\big(\check{C}d^{11\gamma/2+1} + \check{C}d^{-4}\mathbb{E}[|x_0|^{11\gamma+10}]\big)^{1/2}h,
\tag{177}
$$

where $\check{C} := C(\mu, \mu', c, \gamma, \vartheta, L, \mathbb{L}_0, \mathbb{L}_0', \mathbb{L}_1, \mathbb{L}_1')$. Evidently, the desired assertion follows, with

$$
\check{C}(T) := e^{\frac{1}{2}(1+10L+6\mathbb{L}_1)T}, \quad \hat{K}_p := \check{C}d^{11\gamma/2+1}, \quad K_p := \check{C}d^{-4}.
\tag{178}
$$

$\qquad\square$

*Proof of Theorem 2.16.* Using the same technique as the proof of Theorem 2.10, we can obtain the non-asymptotic error bound of pLMC (22). First, by Proposition 2.5, Condition (**A5**) holds true with

$$\mathcal{K}^* = \mathcal{K}, \quad \eta^* = \eta. \tag{179}$$

Using Theorem 3.4 shows

$$\mathcal{W}_2(\nu\check{p}_n, \pi) \leq \hat{\mathcal{K}}_{\check{1}} h + \hat{\mathcal{K}}_{\check{2}} e^{-\lambda nh}, \quad \forall n \in \mathbb{N}_0, \tag{180}$$

where

$$\lambda := \tfrac{\eta}{\log\mathcal{K}+1+\eta/(2L)}, \quad \hat{\mathcal{K}}_{\check{1}} := 2\check{C}(\chi)\Big(\hat{K}_p + C_{\check{1}}K_p d^{(11\gamma+10)/2} + K_p\mathbb{E}[|x_0|^{11\gamma+10}]\Big)^{1/2},$$

$$\chi := \tfrac{\log\mathcal{K}+1}{\eta} + \tfrac{1}{2L}, \quad \hat{\mathcal{K}}_{\check{2}} := e\Big(2(\tfrac{2+2\mu'}{c} + C_{\check{1}})d + 4\mathbb{E}[|x_0|^2]\Big)^{1/2}. \tag{181}$$

In view of Assumption 2.13, one can obtain the desired assertion (25) with

$$\check{C}_1 := Ce^{\frac{1}{2}(1+10L+6\mathbb{L}_1)(\frac{\log\mathcal{K}+1}{\eta}+\frac{1}{2L_1})}, \quad \check{C}_2 := e\Big(2\tfrac{2+2\mu'}{c} + 2C_{\check{1}} + 4\sigma_2(1)\Big)^{1/2}, \tag{182}$$

where $C := C(\mu, \mu', c, \gamma, \vartheta, \sigma_2, L, \mathbb{L}_0, \mathbb{L}'_0, \mathbb{L}_1, \mathbb{L}'_1)$. Thus, the proof is completed. $\square$

# G. Proof of Auxiliary Results for Gaussian Mixture and Double-well Potential

We now verify Assumption 2.8 for the Gaussian mixture. For all $x \in \mathbb{R}^d$, the gradient of $U_1$ is given by

$$\nabla U_1(x) = x - a + \tfrac{2a}{1+e^{2\langle x,a\rangle}}, \tag{183}$$

and the Hessian matrix is given by

$$\nabla^2 U_1(x) = I_d - \tfrac{4e^{2\langle x,a\rangle}}{(1+e^{2\langle x,a\rangle})^2}aa^T. \tag{184}$$

By the definition of the Laplacian operator, we have

$$\Delta U_1(x) = tr\big(\nabla^2 U_1(x)\big) = \sum_{i=1}^d \Big(1 - \tfrac{4a_i^2 e^{2\langle x,a\rangle}}{(1+e^{2\langle x,a\rangle})^2}\Big) = d - \tfrac{4|a|^2 e^{2\langle x,a\rangle}}{(1+e^{2\langle x,a\rangle})^2}. \tag{185}$$

Then

$$\nabla(\Delta U_1(x)) = \tfrac{8|a|^2\big(e^{4\langle x,a\rangle}-e^{2\langle x,a\rangle}\big)}{(1+e^{2\langle x,a\rangle})^3}a = \tfrac{8|a|^2 e^{4\langle x,a\rangle}\big(1-e^{-2\langle x,a\rangle}\big)}{1+3e^{2\langle x,a\rangle}+3e^{4\langle x,a\rangle}+3e^{6\langle x,a\rangle}}a = \tfrac{8|a|^2\big(1-e^{-2\langle x,a\rangle}\big)}{e^{-4\langle x,a\rangle}+3e^{-2\langle x,a\rangle}+3+3e^{2\langle x,a\rangle}}a. \tag{186}$$

By the inequality $1 - e^{-x} \leq x$ and $e^x > 0$, for $x \in \mathbb{R}$, we have, by taking $|a| = 2$,

$$|\nabla(\Delta U_1(x))| \leq \tfrac{8|a|^3|\langle x,a\rangle|}{3} \leq \tfrac{8|a|^4}{3}|x| = \tfrac{128}{3}|x|, \tag{187}$$

as required.

In what follows, we check Assumption 2.14 for the double-well potential. One immediately obtains

$$\nabla U_2(x) = |x|^2 x - x, \quad \nabla^2 U_2(x) = (|x|^2 - 1)I_d + xx^T, \quad \Delta U_2(x) = tr\big(\nabla^2 U_2(x)\big) = (d+1)|x|^2 + d, \tag{188}$$

which further implies

$$\nabla\big(\Delta U_2(x)\big) = (2d+1)x. \tag{189}$$

Thus, we have

$$\big|\nabla\big(\Delta U_2(x)\big)\big| \leq 4d|x|, \tag{190}$$

validating Assumption 2.14.

