# OpenReview forum: "Non-asymptotic Error Bounds in $\mathcal{W}_2$-Distance with Sqrt(d) Dimension Dependence and First Order Convergence for Langevin Monte Carlo beyond Log-Concavity"
_ICML.cc/2025/Conference — ICML 2025 poster_

### Official Review · Reviewer_rDrb · 2025-02-23

**Overall Recommendation:** 3

**Summary:**

When generating samples from a target distribution $\pi$ from a large
dimension $d$ -- including when the normalization constant is unknown -- one
often employs Langevin Monte Carlo (LMC). This method starts by constructing
a Langevin diffusion where its invariant distribution matches the desired
target distribution and then runs a discretized version of the diffusion to
generate samples. Since the discretization stepsize $h$ introduces some
error, these samples are not exactly from the target distribution. This
paper, along with a vast literature before it, aims to quantify the rate of
convergence of LMC samples toward the target distribution in terms of the
$L_2$ Wasserstein metric and terms $h$ and $d$. The paper argues under a
wide set of assumptions, e.g. the target distribution satisfies a
log-Sobolev inequality and dissipaivity condition, that the error rate is
$\tilde{O}(\sqrt{d}h)$ which is state of the art. Crucially, this improves
upon past work that assumes the target distribution must be strongly log
concave. The authors also provide a framework for proving non-asymptotic
results for other samplers, namely the projected LMC sampler pLMC.

## update after rebuttal
Based on the authors feedback and other reviews, I'm inclined to keep my score of a weak accept.

**Claims And Evidence:**

Yes, the simplified theoretical argument in the paper seems reasonable, and
all assumptions required to obtain the desired bounds on the LMC convergence
are clearly laid out. The theoretical arguments are corroborated by some
empirical experiments to demonstrate that their rates of convergence are
followed in the case of a Gaussian mixture model target distribution.

**Essential References Not Discussed:**

To the best of my knowledge (which in this field is not up to date) the
authors have included necessary references.

**Experimental Designs Or Analyses:**

The empirical work in this paper is light, but what the authors have
provided appears valid.

**Methods And Evaluation Criteria:**

The core contribution of the paper is theoretical, although the empirical
experiment offered in Section 4 does align with their theoretical
prescriptions.

**Other Comments Or Suggestions:**

There are a couple possible typos:

L431: It should read "probability distributions"
Figure 1(c) and (d): Should the x-axis be "stepsize" instead of timesteps?
It seems that the discretization error should decrease with the number of
time steps, but perhaps I'm confused by the terminology.

**Other Strengths And Weaknesses:**

For the most part, the paper is very thorough and well written. The
theoretical arguments are carefully laid out and a simplified version in the
main paper offers guidance on how to read the proofs. I found the comparison
to other work also eludicating, as this field has many papers which all have
relatively small but significant differences in the error bounds they can
provide.

The main novelty of the paper appears to be the way that the authors
decompose the problem into two pieces: a finite time mean square fundamental
convergence theorem for SDEs that handles quantifies discretization error
accumulated from LMC, and an appeal to erdogicity that bounds the error from
only running a finite number of LMC steps. This does seem to be a
significant approach for tackling LMC convergence when the target
distribution is appropriately behaved.

This does require that the target distribution satisfy a log-Sobolev
inequality and dissipaivity condition, which may be a larger assumption than
previous work. This may be the work's biggest weakness.

**Questions For Authors:**

[Q1] While most assumptions laid out in this paper are relatively simple to
understand, Assumption 3.3 seems less obvious. How severe of an assumption
is this?

**Relation To Broader Scientific Literature:**

The theoretical work to quantify the convergence rates of LMC is mostly of
theoretical interest, but there is a good chance that ideas from this work
could lead to a practical variation of LMC that achieves better convergence
properties.

**Theoretical Claims:**

I briefly investigated some of the proofs for the arguments supplied
(Appendix B and C) and saw no obvious errors.

---

> ### Author Rebuttal · Authors · 2025-04-01
>
> ## Response to Reviewer rDrb
>
> Thank you for your valuable feedback on our paper. We are grateful for your thoughtful comments, which have guided us in refining the manuscript. Here, we address each of your questions in detail and highlight the changes made accordingly.
>
> ### About *Weakness*
>
> > This does require that the target distribution satisfy a log-Sobolev inequality and dissipaivity condition, which may be a larger assumption than previous work. This may be the work's biggest weakness.
>
> **Response**: Thanks a lot for your valuable comment. The log-Sobolev inequality and dissipaivity condition is indeed more strict than the Talagrand transport inequality and the Poincare inequality. However, the error bound $O(\sqrt{d}h)$ was obtained for LMC under a strongly log-concave condition, see, Li et al. (2022). The main aims of this work are to answer the key question: *Can the error bound $O(\sqrt{d}h)$ still hold true for LMC without the strongly log-concave condition?* As discussed above, compared with the strongly log-concave condition, the log-Sobolev inequality and dissipaivity condition are weak. We aim to do the error analysis under more relaxed conditions for further work.
>
> ### About *Comments Or Suggestions*
>
> > L431: It should read "probability distributions".
>
> **Response**: Corrected! Thanks!
>
> > Figure 1 (c) and (d): Should the x-axis be "stepsize" instead of timesteps? It seems that the discretization error should decrease with the number of time steps, but perhaps I'm confused by the terminology."
>
> **Response**: Thanks for pointing out this issue. You are absolutely right that the x-axis is "stepsize". We are sorry for this and will fix this typo in the revision.
>
> ### About *Questions*
>
> > While most assumptions laid out in this paper are relatively simple to understand, Assumption 3.3 seems less obvious. How severe of an assumption is this?
>
> **Response**: Thanks for your comment. Assumption 3.3 means that moments of a numerical algorithm should be uniform-in-time bounded, which is essentially used in the infinite-time convergence analysis. In Section 2, concrete numerical methods (such as the LMC algorithm and the pLMC algorithm) are provided satisfying Assumption 3.3 under some assumptions (see Lemma 2.9 and Lemma 2.15). In the revision, we will add some comments and discussions following  Assumption 3.3 for better readability.

---

### Official Review · Reviewer_H65o · 2025-03-14

**Overall Recommendation:** 3

**Summary:**

This paper addresses the challenge of sampling from non-log-concave distributions, including those that satisfy a dissipativity condition or a log-Sobolev inequality. The authors approach this problem by discretizing the Langevin dynamics and establish a state-of-the-art convergence rate of d^{1/2}\varepsilon^{-1} in the W_2 distance, under the assumptions of gradient Lipschitz continuity and linear growth of the third derivative. The theoretical findings are further verified by numerical experiments.

**Claims And Evidence:**

The paper’s primary claims are supported by a rigorous theoretical framework and are backed by numerical experiments on controlled examples. However, there are areas where the evidence is less convincing:

- the optimal error bound sounds confusing since this paper gives improved bound with stronger assumptions and there is no lower bound compared to their upper bound.

**Essential References Not Discussed:**

I think the related works appear to be appropriately cited and discussed in the paper.

**Experimental Designs Or Analyses:**

I only check it in high level.

**Methods And Evaluation Criteria:**

The proposed methods are well-suited to the problem. The paper develops a robust uniform-in-time convergence framework and provides optimal error bounds in W_2 distance, which are standard and relevant metrics for evaluating sampling algorithms. The use of synthetic benchmarks, such as Gaussian mixtures and double-well potentials, is appropriate for initial validation, though further testing on diverse or real-world datasets could enhance the evaluation.

**Other Comments Or Suggestions:**

- Line 330, Assumptions 3.1, 3.4, 3.5, 3.3 -> Assumptions 3.1, 3.3, 3.4, 3.5
- Line 331  assssumed -> assumed
- In Assumption 2.6, There exists a dimension independent constant -> There exists a dimension‐independent constant
- Line 424 This framework can also applies to -> This framework can also be applied to

**Other Strengths And Weaknesses:**

There are too many assumptions and the presentation could be clearer.

**Questions For Authors:**

- Can your framework be extended to other metrics?
- Beyond the difference in time horizon, how does your framework in Section 3 differ from that of Tretyakov & Zhang (2013)?

**Relation To Broader Scientific Literature:**

Their framework closely resembles traditional SDE discretization methods. However, while most analyses assume a bounded time horizon, in the context of sampling the relevant time horizon scales as \log d. In this paper, the authors explicitly characterize how the error depends on T.

**Theoretical Claims:**

I only read the proof in the main body.

---

> ### Author Rebuttal · Authors · 2025-04-01
>
> ## Response to Reviewer H65o
>
> We sincerely appreciate your time and effort in reviewing our manuscript. Your insightful comments and constructive suggestions have greatly helped us improve the quality of our work. Below, we provide a point-by-point response to each of your concerns, along with the corresponding revisions in the manuscript.
>
> ### About *Claims and Evidence*
>
> > However, there are areas where the evidence is less convincing: the optimal error bound sounds confusing since this paper gives improved bound with stronger assumptions and there is no lower bound compared to their upper bound.
>
> **Response**: Thank you for pointing this out. We apologize for any confusion caused by this statement. You are right that there is no lower bound here and we will remove the word "optimal" and revise this statement throughout the paper, following your comments.
>
> ### About *Weakness*
>
> > There are too many assumptions and the presentation could be clearer.
>
> **Response**: Thanks. We agree that the current presentation can be improved for better readability. Indeed, Section 3 is a general framework of uniform-in-time convergence for general SDEs and Section 2 is focused on the particular Langevin SDEs. In the revision, we plain to reformulate "assumptions" in Section 3 as several “conditions” (e.g. $H_1$, $H_2$,...). Then Section 2 put some assumptions on Langevin SDEs so that conditions in Section 3 can be satisfied and theoretical results there can be applied to Langevin SDEs. If you have any other good idea, please let us know. Thanks a lot.
>
> ### About *Comments Or Suggestions*
>
> > - Line 330, Assumptions 3.1, 3.4, 3.5, 3.3 -> Assumptions 3.1, 3.3, 3.4, 3.5.
> > - Line 331 assssumed -> assumed.
> > - In Assumption 2.6, There exists a dimension independent constant -> There exists a dimension‐independent constant.
> > - Line 424 This framework can also applies to -> This framework can also be applied to.
> >
> **Response**: Corrected. Thanks!
>
> ### About *Questions*
> >1. Can your framework be extended to other metrics?
>
> **Response**: Thanks for your question. Yes! Our framework can be extended to other metrics, which relied on 3 conditions below:
>
> - the metric satisfies the triangle inequality, such as total variation (TV), $W_p, p\in[1,\infty)$ distances;
> - the underlying Langevin dynamics is exponential ergodic with respect to the chosen metric;
> -  the sampling algorithm  admits finite-time convergence and uniform-in-time moment bounds.
>
> Once these properties are verified, the general framework in Section 3 remains applicable. This would be an interesting direction for our future research.
>
> > 2. Beyond the difference in time horizon, how does your framework in Section 3 differ from that of Tretyakov and Zhang (2013)?
>
> **Response**: Thanks a lot. To be honest, the finite-time convergence theorem (Theorem 3.7) in our paper, as well as Tretyakov and Zhang (2013), follows the original idea of Milstein (1988). The difference is that we need to provide explicit dependence on time and other parameters in the error bound, which are not done in Milstein (1988) and Tretyakov and Zhang (2013). Such explicit dependence, particularly on time $T$, is essential, as it allows us to combine the finite-time error estimates (Theorem 3.7) and the exponential ergodicity of the SDEs to establish the uniform-in-time convergence result (Theorem 3.9). As commented by the reviewer cxoz: "The authors establish this result through a new discretization analysis for SDEs which combines uniform-in-time LMC moment bounds with a finite-time fundamental mean-square convergence theorem".

---

### Official Review · Reviewer_cxoz · 2025-03-17

**Overall Recommendation:** 4

**Summary:**

This paper establishes an almost optimal convergence rate of $\tilde{O}(\sqrt{d}/\epsilon)$ in $W_2$-distance for Langevin Monte Carlo (LMC) when the target measure satisfies the log-Sobolev inequality, along with dissipativity and smoothness conditions. The authors establish this result through a new discretization analysis for SDEs which combines uniform-in-time LMC moment bounds with a finite-time fundamental mean-square convergence theorem. For non-smooth settings where the gradient norm may grow super-linearly, the authors study a projected version of LMC and establish $W_2$ convergence bounds using their discretization analysis.

**Claims And Evidence:**

The claims are supported by rigorous statements and proofs.

**Essential References Not Discussed:**

Most relevant references are cited. There are additional references in the *other comments or suggestions* section below whose discussion can help give a broader picture of LMC analysis. They are presented below.

**Experimental Designs Or Analyses:**

Not applicable since the paper is mostly theoretical.

**Methods And Evaluation Criteria:**

Not applicable.

**Other Comments Or Suggestions:**

* I believe there is an alternative approach to prove Proposition 2.5 which does not rely on Assumption 2.2. Specifically, the log-Sobolev inequality guarantees that $\mathrm{KL}(\nu p_t, \pi) \leq e^{-4t/\rho} \mathrm{KL}(\nu, \pi)$. Moreover, a log-Sobolev inequality implies Talagrand's transport inequality with the same constant, i.e. $W_2(\nu p_t, \pi) \leq \sqrt{\rho \mathrm{KL}(\nu p_t, \pi)}$. Therefore $W_2(\nu p_t, \pi) \leq e^{-2t/\rho} \sqrt{\rho \mathrm{KL}(\nu p_t, \pi)}$, which only required Assumption 2.3.

* Using the above argument, in fact we can weaken Assumption 2.3 to only a Poincaré inequality (as done in Chewi et al., 2024 for LMC) or even to weak Poincaré inequalities that covers heavy-tailed distributions (as done in Mousavi-Hosseini et al., 2023 for LMC). While the transport inequality no longer holds here, $W_2$ can be bound with Rényi distances (see Chewi et al., 2024 and Mousavi-Hosseini et al., 2023 for respective settings). Combined with the discretization analysis here, these approaches can lead to new error bounds for LMC in sub-exponential or heavy-tailed settings.

* For better readability, the authors can change the list of assumptions in line 231 to Assumptions 2.1-2.3 and Assumptions 2.12-2.14.

References:

S. Chewi et al. "Analysis of Langevin Monte Carlo from Poincaré to Log-Sobolev." Foundations of Computational Mathematics 2024.

A. Mousavi-Hosseini et al. "Towards a Complete Analysis of Langevin Monte Carlo: Beyond Poincare Inequality." COLT 2023.

**Other Strengths And Weaknesses:**

**Strenghts**:

As mentioned above, the discretization analysis can open the room for novel analyses of LMC under different assumptions or for analyzing different variants of LMC. The paper also handles locally Lipschitz potentials which is a valuable contribution that has received less attention in the literature.

**Weaknesses**:

* There is not sufficient discussion on the implications of Assumption 2.2, and I think certain statements about when it holds might be incorrect. In fact, this assumption may even not he necessary, I think there are alternative approaches to prove continuous-time exponential ergodicity in $W_2$ under a log-Sobolev inequality, discussed below.

* The dependence on most constants is implicit, which makes interpreting the results for certain examples complicated. The dimension-dependence of some constants is also not clear.

**Questions For Authors:**

1. I don’t see how Assumption 2.2 follows from Assumption 2.6. A simple application of Cauchy-Schwartz and triangle inequalities results in
$$\langle x - y, \nabla U(x) - \nabla U(y)\rangle \geq -(2L’_1 d^{1/2} + L_1 \vert x \vert + L_1 \vert y \vert)\vert x - y\vert$$
But $\vert x \vert, \vert y \vert$ are not bounded. Also, dependence on the initial Wasserstein distance is not clear.

    It seems more like Assumption 2.2 is a form of relaxed convexity. For $L = 0$, it exactly implies convexity of negative log-density. There should be a discussion on when this assumption is satisfied.

2. The LSI constant $\rho$ should be missing somewhere in Proposition 2.5 and Theorems 2.10 and 2.16, as it controls the convergence rate.

3. How does $C_\nabla$ of Lemma 2.15 depend on dimension?

**Relation To Broader Scientific Literature:**

The analysis of Langevin Monte Carlo is a fundamental problem in sampling and of interest to many researchers in the ICML community. Beyond establishing the optimal $W_2$ convergence rate for LMC under the log-Sobolev inequality and smoothness, the discretization analysis introduced here may be used in future analyses of LMC and its variants as well.

**Theoretical Claims:**

I didn't verify the correctness of the proofs, but the overall strategy seems sound.

---

> ### Author Rebuttal · Authors · 2025-04-01
>
> ## Response to Reviewer cxoz
>
> We really appreciate your carefully reading and insightful comments. We will respond to each comment below and revise the manuscript according to these suggestions.
>
> ### About *Weakness*
>
> >1. There is not sufficient discussion on the implications of Assumption 2.2, …, discussed below.
>
> **Response**: Thanks. You are absolutely right. We apologize for your confusion caused by a typo here. The correct condition in Assumption 2.2 should be $$\langle x-y,\nabla U(x)-\nabla U(y)\rangle\geq -L|x-y|^2.$$ We confirm that the analysis throughout the paper is based on the correct form above, and we will fix this typo in the revision.
>
> >2. The dependence on most constants is implicit, …. The dimension-dependence of some constants is also not clear.
>
> **Response**: Thank you. We apologize for any confusion caused by the unclear of dependence to the constants. We will discuss the dependencies of the parameters in detail and explicitly explain how they depend on the number of dimensions $d$ in the revision.
>
> ### About *Comments or Suggestions*
> >1. I believe there is an alternative approach to prove Proposition 2.5 which does not rely on Assumption 2.2. …, which only required Assumption 2.3.
> >2. Using the above argument, …, these approaches can lead to new error bounds for LMC in sub-exponential or heavy-tailed settings.
>
> **Response**: Thanks for your insightful and interesting discussion. The direction you suggested is really promising for extending the analysis to more general settings. However, essential difficulties still exist. On the one hand, we believe Proposition 2.5 can hold without Assumption 2.2, but the analysis of finite-time mean-square convergence of numerical methods with convergence rates essentially rely on the use of Assumption 2.2, which is nothing but the one-sided Lipschitz (also called monotonicity) condition on the drift coefficients $-\nabla U(x)$. As far as we know, without the monotonicity condition, obtaining mean-square convergence rates is highly non-trivial for numerical SDEs (see Hutzenthaler and Jentzen, Ann. Probab., 2020). On the other hand, our approach relies on two key properties: (i) the chosen metric must satisfy the triangle inequality; (ii) the metrics on both sides of the inequality must remain consistent. Since the KL divergence does not satisfy the triangle inequality, inequalities such as KL$(\nu p_t,\pi)\leq e^{-4t/\rho}$KL$(\nu,\pi)$ do not work seemingly.
>
> We sincerely thank the reviewer for bringing these two works of Chewi et al. (2024) and Mousavi-Hosseini et al. (2023) into our notice. Ideas in these works are very inspiring, and we will definitely aim to extend our framework to incorporate weaker functional inequalities and alternative metrics  for our future research. This is not trivial and we need more time to do it.
> In the revision, we will cite and make some comments on these two papers.
>
> >3. For better readability, the authors can change the list of assumptions in line 231 to Assumptions 2.1-2.3 and Assumptions 2.12-2.14.
>
> **Response**: Thanks for your suggestions. We will do this in the revision.
>
> ### About *Questions*
> >1. I don’t see how Assumption 2.2 follows from Assumption 2.6. …. There should be a discussion on when this assumption is satisfied.
>
> **Response**: Thanks a lot for your helpful comments. As mentioned to the first question in *About the weakness*, this issue comes from a typo in the previous manuscript, which will be corrected in the revision.
>
> >2. The LSI constant $\rho$ should be missing somewhere in Proposition 2.5 and Theorems 2.10 and 2.16, as it controls the convergence rate.
>
> **Response**: Thanks a lot for your constructive suggestions. In the revision, we will explicitly show it in corresponding statements.
>
> >3. How does $C_{\nabla}$ of Lemma 2.15 depend on dimension?
>
> **Response**: Thanks a lot for pointing out this issue. The constant $C_{\nabla}$ is independent of the dimension, as it can be written $C_{\nabla}=L_1'+L_1$. Please refer to Lemma 3.3 in Pang et al.(2025) for details.

---

### Official Review · Reviewer_DL9W · 2025-03-21

**Overall Recommendation:** 3

**Summary:**

The authors derive a sampling error bound in Wasserstein-2 for a discrete time discretization of Langevin Monte Carlo. The bound contains two parts, an error term due to finite time truncation of the Langevin dynamics, and an error term due to discretization over a finite time horizon. The innovation of this work appears to be in the analysis of the finite time discretization term, which is done entirely through moment-based calculations. Notably, they bound the error accumulation over short time intervals (Lemma E.1) and long time intervals (Lemma E.2) using rather many assumptions on the regularity of the drift, plus dissipativity of the target measure.

The dissapitivity condition implies some uniform-in-time moment estimates (Lemma 2.4) on the true and discretized processes, and the regularity assumptions on the drift are used to transfer these moment estimates into bounds on the discretization error over short time intervals.

This analysis is similar in style to bounds on ODE discretization error, where $O(h)$ step size dependence is typical. However, to carry out this approach for stochastic dynamics the authors must apply Itô's formula to the drift $\nabla U(x)$, forcing them to use Assumption 2.8 linear growth of $|\nabla(\Delta U(x))|$ which appears to be uncommon in the literature.

**Claims And Evidence:**

The theoretical claims made in this work are supported by clear and convincing evidence in the form of proofs.

**Essential References Not Discussed:**

Please discuss how the proof in this work differs from that of Li et al. 2022. How (if at all) is the log-sobolev condition used in the discretization error bound over a finite time horizon? I ask because, it is important to clarify which parts of the present work are original relative to Li et al. 2022, which uses similar assumptions and a similar technique (Gronwall-based discretization error bounds via direct control of moments). Does the finite time discretization error analysis of Li et al. 2022 make use of log-concavity? If not, would their approach work equally well using only a log-sobolev inequality to bound error due to truncation at a finite time?

**Experimental Designs Or Analyses:**

Figures 1(c) and 1(d) are a convincing proof of concept for this work.

**Methods And Evaluation Criteria:**

N/a

**Other Comments Or Suggestions:**

N/a

**Other Strengths And Weaknesses:**

The major weakness of this work is that it is rather unclear what is the originality, given the significant similarities between it and Li et al. 2022. Relaxing the assumptions of Li et al. 2022 from log-concavity to merely log-sobolev is rather incremental if the proof techniques are largely the same. Another weakness of this work is that it is hard to read because of many different assumptions introduced throughout. I counted 9 assumptions spread across Section 2, only 5 of which are required by Theorem 2.10, but then four more assumptions stated in Section 3, which contains Theorem 3.7 that is essential in the proof of Theorem 2.10. Does Theorem 2.10 also require the assumptions in Section 3? Are any of these assumptions redundant? Could the presentation be simplified so it is easier to keep track of the many requirements of this analysis?

**Questions For Authors:**

1. Does Theorem 2.10 require Assumptions 3.1, 3.3, 3.4, 3.5 indirectly through its use of Theorem 3.7? If so, please add them to the statement of Theorem 2.10 so that the statement is correct. Are any of the assumptions stated in Section 3 redundant?
2. How does the proof technique in this work differ from that of Li et al. 2022? What are the original contributions contained in the techniques used by the present paper?

**Relation To Broader Scientific Literature:**

According to Table 1, this work proves an optimal error bound of $\tilde{O}(d^{1/2} \epsilon^{-1})$ without log-concavity. A bound of the same order was already shown in (Li et al. 2022) using both log-concavity and the third order growth condition on $U(x)$ required by this work.

**Theoretical Claims:**

I read the proofs of Theorem 2.1, Lemma 2.4, Theorem 3.7, Lemma E.1, and Lemma E.2.

---

> ### Author Rebuttal · Authors · 2025-04-01
>
> ## Response to Reviewer DL9W
>
> We sincerely thank the reviewer for constructive suggestions and comments. Next we address all comments point-by-point and  will revise the manuscript to incorporate these suggestions.
>
> ### About *Summary*
> > However, to carry out … Assumption 2.8 … which appears to be uncommon in the literature.
>
> **Response**: Thanks. We would like to mention that, Assumption 2.8 comes from Li et al. 2022, where the authors remarked that, it is not necessarily stronger than the widely used Hessian Lipschitz condition (see Section 4 in Li et al. 2022). In fact, as pointed out by Li et al. 2022, there exist potentials, e.g. $U(x)=x^{4}$, that satisfy Assumption 2.8 but violate the Hessian Lipschitz condition. Moreover, it is shown that a class of Gaussian mixtures satisfy Assumption 2.8.
>
> ### About *Essential References Not Discussed*
> >1. Please discuss how the proof in this work differs from that of Li et al. 2022.
>
> **Response**: Thanks. To show the difference between our work and Li et al. (2022) clearly, it is worthwhile to illustrate the main idea of Li et al. (2022). Indeed, their arguments are based on a direct long-time mean-square convergence analysis of the numerical method, which essentially relies on the use of the log-concavity condition. To see this fact, we provide a key idea behind their error analysis. Suppose we get the following error estimates:
> $$E[|X_{t_{k+1}}-Y_{k+1}|^2]\leq E[(1+\epsilon h)|X_{t_k}-Y_k|^2-2h\langle X_{t_k}-Y_k , \nabla U(X_{t_k})-\nabla U(Y_k)\rangle]+c_2 h^3,
> $$ where $\epsilon >0$ can be sufficiently small. Then the log-concavity condition, i.e. $$\langle x-y,\nabla U(x)-\nabla U(y)\rangle \geq c_1 |x-y|^2, c_1>0
> $$ is essentially used here to arrive at the contraction: $$E[|X_{t_{k+1}}-Y_{k+1}|^2]\leq[(1-(2c_1-\epsilon)h]E[|X_{t_{k}}-Y_{k}|^2]+c_2h^3,
> $$ for $0<\epsilon<2c_1$. Armed with the contraction, one can easily get the uniform-in-time error bound by iteration. However, without the log-concavity, this framework does not work in obtaining uniform-in-time error bounds.
>
>  In contrast to Li et al. (2022), we develop a new framework of uniform-in-time error analysis without the log-concavity. The proof consists of two key components. First, we derive the finite-time mean-square convergence error bounds for the LMC, which grow exponentially with respect to the time length $T$. Arguments for this step require only Lipschitz condition and avoid any convexity assumption. Second, we obtain uniform-in-time error bounds by relying on the exponential ergodicity of the Langevin dynamics, which is available under one-side Lipschitz condition and log-Sobolev inequality (see Subsection 2.4 for details). Moreover, our framework also works for the case of non-globally Lipschitz continuous $\nabla U$, which is not even investigated in Li et al. (2022).
>
> To summarize, the approach of error analysis is essentially different from Li et al. (2022) and more relaxed conditions (non log-concavity and non-glabally Lipschitz conditions) are used to cover more problems.
>
> >2. How (if at all) is the log-sobolev condition … due to truncation at a finite time?
>
> **Response**: Thanks. The LSI is not used in the finite-time error analysis of our paper. It is only used/required to obtain the exponential ergodicity of the Langevin dynamics (see Proposition 2.5 and Step 3 in Subsection 2.4 for details). As explained above, the authors of Li et al. (2022) did not do finite-time error analysis but used the log-concavity essentially to directly carry out a long-time error analysis and derive an infinite-time convergence theorem.
>
> ### About *Weakness*
> >1. The major weakness … are largely the same.
>
> **Response**: We deeply apologize for not making the originality clear, to make you confused. As explained in our response to your previous comments, the approach of error analysis in our paper is essentially different from Li et al. (2022). As commented by the reviewer cxoz: "The authors establish this result through a new discretization analysis for SDEs which combines uniform-in-time LMC moment bounds with a finite-time fundamental mean-square convergence theorem".
>
> >2. Another weakness … of this analysis?
>
> **Response**: : Thanks. We apologize for the current presentation.
> Indeed, Section 3 is a general framework of uniform-in-time convergence for discretizations of general SDEs, independent of Section 2. So Theorem 2.10 does not use assumptions in Section 3.
> Some assumptions in Section 2 for particular Langevin SDEs can be regarded as particular ones in Section 3. But they are not redundant and we agree that the current presentation can be improved in the revision for better readability. Due to the work limitation, please refer to our responses to *About weakness* of the reviewer H65o for details.
>
> ### About *Questions*
> **Response**: Thanks. These concerns have been carefully addressed previously. Hope you are satisfied with our answers. Please tell us once you have further questions.

---

> > ### Comment · Reviewer_DL9W · 2025-04-03
> >
> > I thank the authors for the informative and clarifying responses to my questions. I especially appreciate the very clear explanation how this work differs from the approach taken by Li et al. 2022. Conditional on the proposed changes (see below) to make the assumptions clearer, I am willing to raise my score to 3.
> >
> > Proposed changes:
> > " In the revision, we plain to reformulate "assumptions" in Section 3 as several “conditions” (e.g,...). Then Section 2 put some assumptions on Langevin SDEs so that conditions in Section 3 can be satisfied and theoretical results there can be applied to Langevin SDEs. "

---

> > > ### Author Response · Authors · 2025-04-07
> > >
> > > Thank you so much for your raising the score to 3. As you proposed, we will make some changes in the revision. Again, thanks for your help and suggestions.

---

### Decision · Program_Chairs · 2025-05-01

**Decision:**

Accept (poster)

**Comment:**

This paper analyzes the convergence of Langevin Monte Carlo. There are several contributions: 1) convergence in W2 under LSI instead of commonly assumed, stronger assumption of log-concavity; 2) sqrt(d) dimension dependence, which was what people tried for a long time and a substantial progress in the literature, however established under stronger conditions; 3) a proof that quantitatively shows that a recent modification of LMC (Pang+ 25) works under superlinear growth condition instead of global Lipschitzness, which is weaker. All reviewers and I agree this is a good paper. Although no machine learning application has been discussed, this is a work on the theoretical foundation of machine learning. I hence recommend acceptance as poster (not higher due to insufficient machine learning components), however under the condition that the authors remove the overclaimed word "optimal" as some reviewer pointed out (and I agree).